# A whole-brain connectivity map of mouse insular cortex

**Daniel A Gehrlach[1,2†‡], Caroline Weiand[1,3†], Thomas N Gaitanos[1], Eunjae Cho[1], Alexandra S Klein[1,2], Alexandru A Hennrich[4], Karl-Klaus Conzelmann[4], Nadine Gogolla[1]\***

[1]Max Planck Institute of Neurobiology, Circuits for Emotion Research Group, Martinsried, Germany; [2]International Max-Planck Research School for Molecular Life Sciences, Munich, Germany; [3]International Max-Planck Research School for Translational Psychiatry, Munich, Germany; [4]Max von Pettenkofer-Institute and Gene Center, Medical Faculty, Ludwig-Maximilians-University Munich, Munich, Germany

**Abstract** The insular cortex (IC) plays key roles in emotional and regulatory brain functions and is affected across psychiatric diseases. However, the brain-wide connections of the mouse IC have not been comprehensively mapped. Here, we traced the whole-brain inputs and outputs of the mouse IC across its rostro-caudal extent. We employed cell-type-specific monosynaptic rabies virus tracings to characterize afferent connections onto either excitatory or inhibitory IC neurons, and adeno-associated viral tracings to label excitatory efferent axons. While the connectivity between the IC and other cortical regions was highly bidirectional, the IC connectivity with subcortical structures was often unidirectional, revealing prominent cortical-to-subcortical or subcortical-to-cortical pathways. The posterior and medial IC exhibited resembling connectivity patterns, while the anterior IC connectivity was distinct, suggesting two major functional compartments. Our results provide insights into the anatomical architecture of the mouse IC and thus a structural basis to guide investigations into its complex functions.

**\*For correspondence:**
ngogolla@neuro.mpg.de

[†]These authors contributed equally to this work

**Present address:** [‡]HMNC Brain Health, Wilhelm-Wagenfeld-Strasse, Munich, Germany

**Competing interests:** The authors declare that no competing interests exist.

## Introduction

The insular cortex (IC or insula) has been suggested to mediate a wide variety of brain functions, such as the processing of external and bodily sensory information (*Gogolla, 2017*; *Kurth et al., 2010*), bodily- and self-awareness (*Craig, 2009*; *Craig, 2011*), emotion regulation (*Etkin et al., 2015*), feelings and complex social-affective functions like empathy (*Damasio and Carvalho, 2013*), and switches between large-scale brain networks (*Menon and Uddin, 2010*).

Rodent studies further demonstrated roles for the IC in multisensory (*Gogolla et al., 2014*; *Rodgers et al., 2008*) and pain processing (*Tan et al., 2017*), representation of valence (*Wang et al., 2018*), learning and memory (*Bermudez-Rattoni et al., 2005*; *Lavi et al., 2018*), social interactions (*Rogers-Carter et al., 2018*), gustation (*Peng et al., 2015*; *Wang et al., 2018*), drug cravings and malaise (*Contreras et al., 2007*), and aversive states such as hunger, thirst, and anxiety (*Gehrlach et al., 2019*; *Livneh et al., 2020*; *Livneh et al., 2017*).

While anatomical studies in diverse species highlight that the insula is one of the most complex anatomical hubs in the mammalian brain (*Allen et al., 1991*; *Cauda et al., 2012*; *Cechetto and Saper, 1987*; *Menon and Uddin, 2010*; *Yasui et al., 1991*), to date, there is no comprehensive connectivity map of the IC of the mouse, a genetically accessible model organism widely employed in systems neurosciences.

Here, we aimed at providing a comprehensive input and output connectivity description of the mouse IC to facilitate the mechanistic investigation of insula functions. Furthermore, we compared

the connectivity structure of the IC along its rostro-caudal axis to establish a connectivity-based compartmentalization that may facilitate the comparison across species. Indeed, most physiological and functional studies target specific subregions, often referred to as aIC and pIC, without clear consensus on boarders and coordinates of these regions. Toward the goal of providing a connectivity-based structure to future functional studies, we divided the mouse IC into three equally large subregions along its rostro-caudal extent, namely an anterior, medial, and posterior insular part (aIC, mIC, and pIC, respectively) spanning its entire extent. Although connectivity differences between granular (GI), dysgranular (DI) or agranular (AI) parts of the IC have been reported previously (*Maffei et al., 2012*), we did not distinguish them here, due to the technical challenge of specifically targeting these subdivisions. Instead, we focus on cell-type-specific monosynaptic retrograde rabies virus tracings (*Wickersham et al., 2007a*) to separately map inputs to excitatory and inhibitory neurons of the IC across all of its subregions. To label outputs, we performed axonal AAV labeling of excitatory efferents of the aIC, mIC and pIC.

We provide a whole-brain analysis of bidirectional connectivity of the longitudinal IC subregions for the two major neuronal subclasses that is excitatory pyramidal neurons and inhibitory interneurons.

## Results

### Viral-tracing approach to reveal the input-output connectivity of the mouse IC

To map the connectivity of the entire mouse IC, we injected viral tracers into three evenly spaced locations along the rostro-caudal axis with the aim of comprehensively tracing from its entire extent and to assess possible parcellation of the mouse IC into connectivity-based subdomains. The most anterior region, aIC, ranged from +2.45 mm to +1.20 mm from Bregma; the medial part, mIC, from +1.20 mm to +0.01 mm from Bregma, and the posterior part, pIC, from +0.01 mm to −1.22 mm from Bregma (see also *Figure 1C*).

In order to trace the monosynaptic inputs to the IC, we utilized a modified SADΔG-eGFP(EnvA) rabies virus (RV), which has been shown to label monosynaptic inputs to selected starter cells with high specificity (*Wall et al., 2010*; *Wickersham et al., 2007b*). This virus lacks the genes coding for the rabies virus glycoprotein (G) and is pseudotyped with the avian viral envelope EnvA. This restricts its infection to neurons expressing the avian TVA receptor and to monosynaptic retrograde infection of afferents (*Figure 1A*). We infected the IC of *Camk2a*-Cre and *Gad2*-Cre expressing mouse lines to specifically target TVA and rabies virus glycoprotein expression to excitatory pyramidal or inhibitory interneurons, respectively (see *Figure 1A* and Materials and methods).

In order to trace and quantify the axonal projections (outputs) of the IC, we injected Cre-dependent adenoassociated virus (AAV2/5-DIO-eYFP) into *Camk2a*-Cre and *Gad2*-Cre transgenic mice (see *Figure 1B* and Materials and methods). We did not observe long-range projections from IC *Gad2*-Cre tracings (data not shown). Therefore, we here only present outputs from excitatory projection neurons performed in *Camk2a*-Cre transgenic mice.

We first confirmed the reliability and accuracy of a semi-automated approach that we developed to quantify starter cells and input cells of rabies tracings. We compared counts from three different humans and also human versus automated counts. We then calculated the relative percent difference (RPD) for comparisons between three different humans counting RV-labeled input cells within the same brain and found that RPDs ranged from 0.64% to 1.08%. Comparing human versus automated cell counts for the same brain resulted in RPDs ranging from 0.02% to 0.67%, while comparing human versus automated counts in three different brains resulted in RPDs ranging from 0% to 2% (see Materials and methods and *Figure 1—figure supplement 3* for details). The overall low RPDs validated our semi-automated approach.

We next assessed the spread and quality of our starter cell populations for both AAV and RV tracings in a semi-automated manner (*Figure 1C*, *Figure 1—figure supplement 1*, *Figure 1—figure supplement 2* and Materials and methods). For RV experiments, starter cells were counted when double positive for eGFP (i.e. SADΔG$^+$) and mCherry (i.e TVA$^+$). For AAV tracings, starter cells were counted as eYFP-positive cell bodies. We thereby defined both the total number and location of all starter cells. The bulk of the starter populations for the distinct IC subregions were highly separated

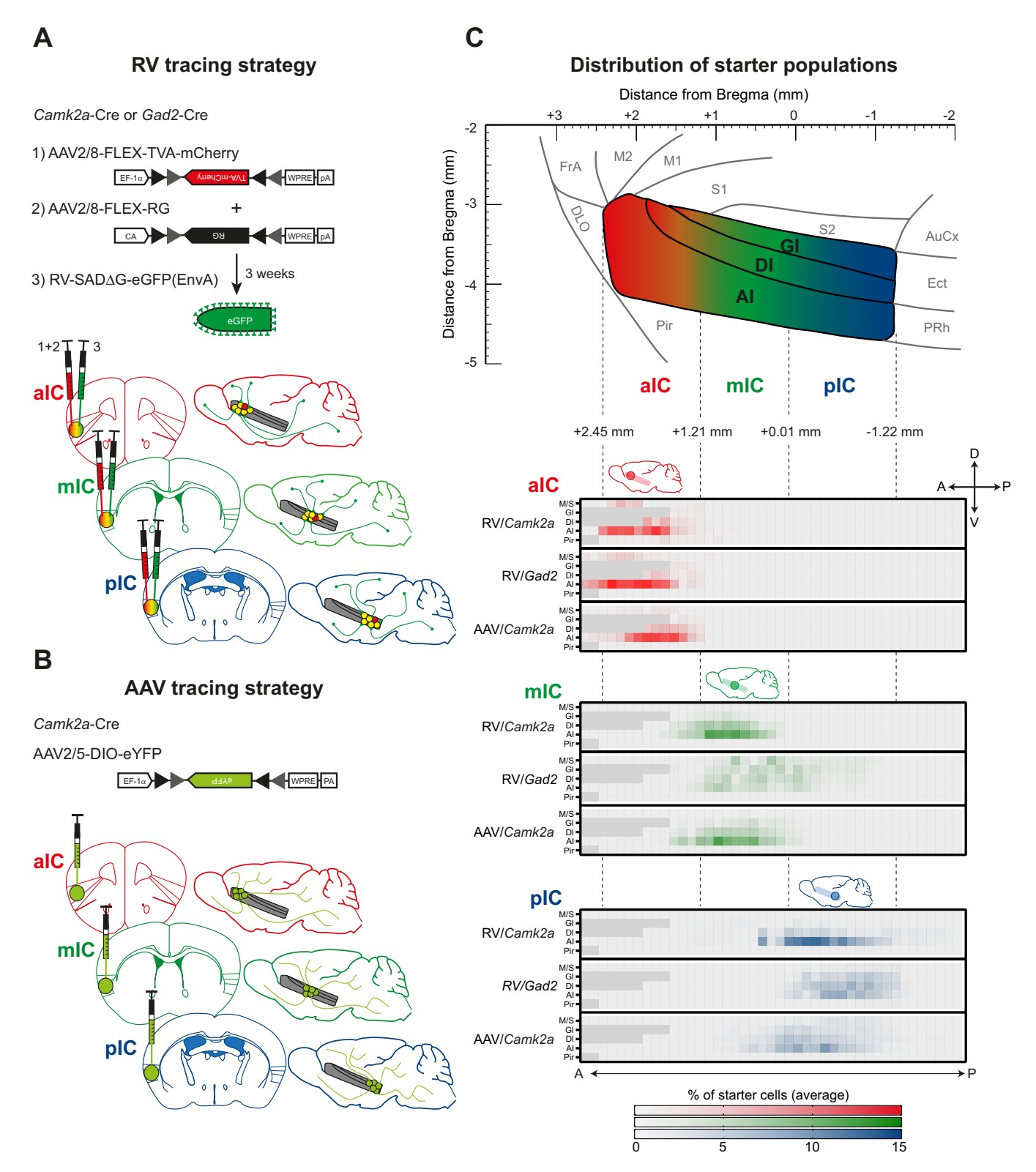

**Figure 1.** Tracing strategy and localizations for input and output viral tracings from distinct IC subregions. Schematic representation of Cre-dependent (**A**) monosynaptic retrograde Rabies virus tracing (RV) and (**B**) anterograde axonal AAV tracings (AAV) used to determine respective input and output connectivity to the IC. Tracings were performed in both excitatory (*Camk2a*-Cre) and inhibitory (*Gad2*-Cre) mouse lines for RV, and only in the *Camk2a*-Cre mouse line for AAV. For RV tracings, AAV-FLEX helper viruses expressing mCherry-tagged TVA (1) and rabies-virus-specific G protein (2) were co-

*Figure 1 continued on next page*

*Figure 1 continued*

injected into the IC region of interest. Three weeks later EnvA-coated, eGFP-expressing modified RV lacking G protein was injected at the same location (3). For anterograde tracings, a one-off injection of eYFP-expressing AAV-FLEX virus was administered into the chosen location. Three distinct IC subregions were chosen for each tracing technique: anterior (aIC, red), medial (mIC, green) or posterior (pIC, blue). (C) Schematic illustration of the lateral view of the IC including distances from Bregma (top panel) and heatmap showing average starter cell distribution for each tracing strategy at each specific IC target (bottom panels). The three IC target subregions were mostly non-overlapping, and only a minimal percentage of cells were detected in the Motor and Sensory Cortex (M/S), or Piriform Cortex (Pir) neighboring the IC. n = 3 mice per injection site/tracing strategy. Heatmap intensity scale is the same for all three IC target subregions. Regions absent at specific Bregma levels indicated by dark gray squares.

The online version of this article includes the following figure supplement(s) for figure 1:

**Figure supplement 1.** Starter cell identification.
**Figure supplement 2.** Starter cells.
**Figure supplement 3.** Cell counting.
**Figure supplement 4.** Workflow of tracing quantification and data analysis.

and non-overlapping for both RV and AAV tracings (*Figure 1C*, *Figure 1—figure supplement 1C*). In some cases, a small percentage of starter neurons were detected in regions outside the boundaries of the IC, including in the Pir, S1 and S2, as well as the M1 (*Figure 1—figure supplement 2A, C*). In these cases, we asked if contamination affected the qualitative connectivity structure. Toward this goal, the percentage of total input was compared between tracings with different degrees of spillover. There was no consistent qualitative difference in connectivity patterns due to strength of spillover identifiable between the brains used in this study (see Materials and methods, Discussion and *Figure 1—figure supplement 2D*). However, we excluded brains where separate starter cell populations were detected outside the IC (see example picture *Figure 1—figure supplement 2A*, right) or that did not yield strong starter cell populations (see also Materials and methods for exclusion criteria). We also analyzed the layer distribution of our starter cells. While we found starter cells across all cortical layers, there were substantial differences in starter cell distribution across layers. However, the layers with more or less starter cells were consistent across Cre-lines and viruses employed, and may thus reflect more general anatomical features that cannot be attributed to the Cre-lines or viruses employed (see *Figure 1—figure supplement 2B*).

To compile the whole-brain connectivity maps for both RV and AAV tracings, we cut coronal sections (ranging from +2.65 to −6.2 mm relative to Bregma) and analyzed the long range, ipsilateral connectivity on sections approximately 140 μm apart (*Figure 1—figure supplement 4A–C*, Materials and methods). For the rabies virus tracings, we obtained brain-wide inputs ranging from 5000 to 45,000 cells, with convergence ratios ranging from 6 to 15 (*Figure 1—figure supplement 1C*). We accounted for the variability between tracings by normalizing cell counts per region of interest (ROI) to the total number of neurons per brain. We additionally obtained the cell density of each ROI as cells/mm$^2$.

For the AAV tracings, we identified a total of 600–800 million pixels per brain as IC efferents (*Figure 1—figure supplement 1C*). The tracings yielded a combination of identifiable single axons and dense axon bundles, where we could not discriminate between individual axonal fibers. Therefore, output from IC was quantified as overall amount of eYFP-positive pixels (and not axon numbers) within distinct target brain regions. To account for animal-to-animal variation, we normalized each ROI to the total amount of pixels identified brain-wide. Additionally, we calculated the innervation density, given in percent of maximal ROI pixel count.

For both RV and AAV tracings, we determined the spatial location of the starter neurons. Within this immediate surround of the starter cells we did not quantify inputs or outputs due to the ambiguity to distinguish starter cells from input or outputs, respectively (*Figure 1C*, *Figure 1—figure supplement 4D*). Thus, the quantifications of this study focus on the long-range connectivity of IC subregions.

To ensure Cre-dependence of our approach, we performed control infections of WT mouse brains. Mice lacking Cre-recombinase should not express eGFP when infected with RV. Indeed only some GFP+ neurons were detected at the injection sites within the boundaries that we would normally exclude from our quantitative analysis (*Figure 1—figure supplement 4D*). To test the dependence on RG supplementation for the synaptic jump of the virus and thus to ensure the monosynaptic restriction, we injected TVA and RV into *Camk2a*-Cre or *Gad2*-Cre mice without the

addition of RG. As expected, eGFP expression was detected in transfected neurons, but none was expressed outside the boundaries that we would normally exclude from our quantitative analysis, indicating that no synaptic jump had occurred and no long-range projections were labeled (*Figure 1—figure supplement 4D*).

## Whole brain input/output map of mouse IC

To provide a detailed account of the brain-wide connectivity of the mouse IC, we analyzed its bidirectional connectivity with 75 anatomical subregions (the detailed connectivity maps of the IC with all subregions analyzed can be found in the *Figure 2—figure supplements 1–3*). To first gain an overview of the overall IC connectivity, we pooled these detailed datasets into overall connectivity patterns between the IC and 17 larger brain regions (*Figure 2*).

While there were some quantitative differences, overall the anterior to posterior extent of the IC connected to largely the same major brain regions and no major brain region was exclusively connected to one but not the other IC regions. We also did not observe marked differences in the connectivity patterns of inhibitory versus excitatory neurons, as both major neuronal cell classes exhibited similar connectivity patterns. However, while both, excitatory and inhibitory cells of all IC subregions received strong intrainsular inputs and inputs from sensory cortices, ordinary one way-ANOVAs, performed for each brain region separately, revealed that specifically the prefrontal ($F_{(2,6)}$ = 7.610, *p=0.0226), motor ($F_{(2,6)}$ = 8.586, *p=0.0174) and association cortices ($F_{(2,6)}$ = 22.16, *p=0.0017) sent significantly different amounts of inputs onto inhibitory neurons of the aIC, the mIC or pIC (*Figure 2A*, top left; see also *Supplementary file 3* for statistics). Tuckey's posthoc multiple comparisons test showed that aIC received stronger inputs from the prefrontal cortex compared to mIC and pIC (aIC vs. mIC *p=0.0413 and aIC vs. pIC *p=0.0413) and from the motor cortex compared to pIC (*p=0.0185). Associative cortices on the other hand sent stronger inputs onto pIC inhibitory neurons than onto aIC or mIC inhibitory neurons (pIC vs. aIC **p=0.0017 and pIC vs. mIC *p=0.0141). Inputs to mIC were stronger compared to inputs to the aIC (*p=0.0407).

Concerning the projections arising from the IC, a one-way ANOVA found a difference in output strength toward other IC regions ($F_{(2,6)}$ = 0.5403, *p=0.0455). Posthoc testing showed that the pIC projected more strongly to other IC regions than the aIC (*p=0.0498; *Figure 2A*, top right; see also *Supplementary file 3* for statistics).

We next assessed the connectivity of the IC with subcortical brain regions. Overall the IC connectivity was characterized by three major connections: strong projections to the striatum from the aIC, and bidirectional connections with diverse subregions of the amygdala and the thalamus. One-way ANOVAs further revealed significant differences between excitatory inputs to olfactory areas ($F_{(2,6)}$ = 6.360, *p=0.0329) and the pallidum ($F_{(2,6)}$ = 5.147, *p=0.0499), as well as inhibitory inputs to the amygdala ($F_{(2,6)}$ = 15.80, **p=0.0041) and excitatory outputs to the claustrum ($F_{(2,6)}$ = 45.59, ***p=0.0002), the striatum ($F_{(2,6)}$ = 10.13, *p=0.0119) and the hindbrain ($F_{(2,6)}$ = 9.178, *p=0.0149). Posthoc Tuckey's multiple comparison tests showed that excitatory inputs from olfactory areas, were significantly stronger to the mIC than to the aIC (*p=0.0353). Further, the inhibitory neurons of the pIC received significantly stronger inputs from the amygdala than the mIC (*p=0.0443) or aIC (**p=0.0033). Concerning outputs, projections to the striatum were significantly stronger from the aIC than from the mIC (*p=0.0194) or the pIC (*p=0.0204; *Figure 2A*, bottom; see also *Supplementary file 3* for statistics). Interestingly, the mIC, containing the 'gustatory cortex' was most heavily connected with olfactory regions and sent more projections to the claustrum than pIC (***p=0.0002) or aIC (**p=0.004). However, also the pIC sent more projections to the claustrum than aIC (**p=0.005). Furthermore, the aIC sent significantly more excitatory outputs to the Hindbrain than the pIC (*p=0.0125). Between IC subregions along the rostro-caudal axis, we found that, overall, the pIC received twice as many inputs from the sensory cortices (41 ± 11% of total excitatory input connectivity) as the other IC subregions (20 ± 5% and 23 ± 7% for mIC and aIC, respectively). In contrast, the aIC received the majority of inputs from the motor cortex.

The aIC sent almost one third of its projections to the striatum, while for the mIC and pIC about 10% of the efferents were innervating the striatum. (aIC 32 ± 6% of outputs, as compared to 9 ± 1% for mIC and 11 ± 2% for the pIC). An inverse pattern was observed for the amygdala projections. About 5–7% of the mIC's and pIC's efferents were directed to different amygdala subnuclei, while only 1.5% of the aIC efferents were directed to the amygdaloid complex. Given the strong connectivity of the entire IC with important subcortical regions, such as the striatum, the amygdala or the

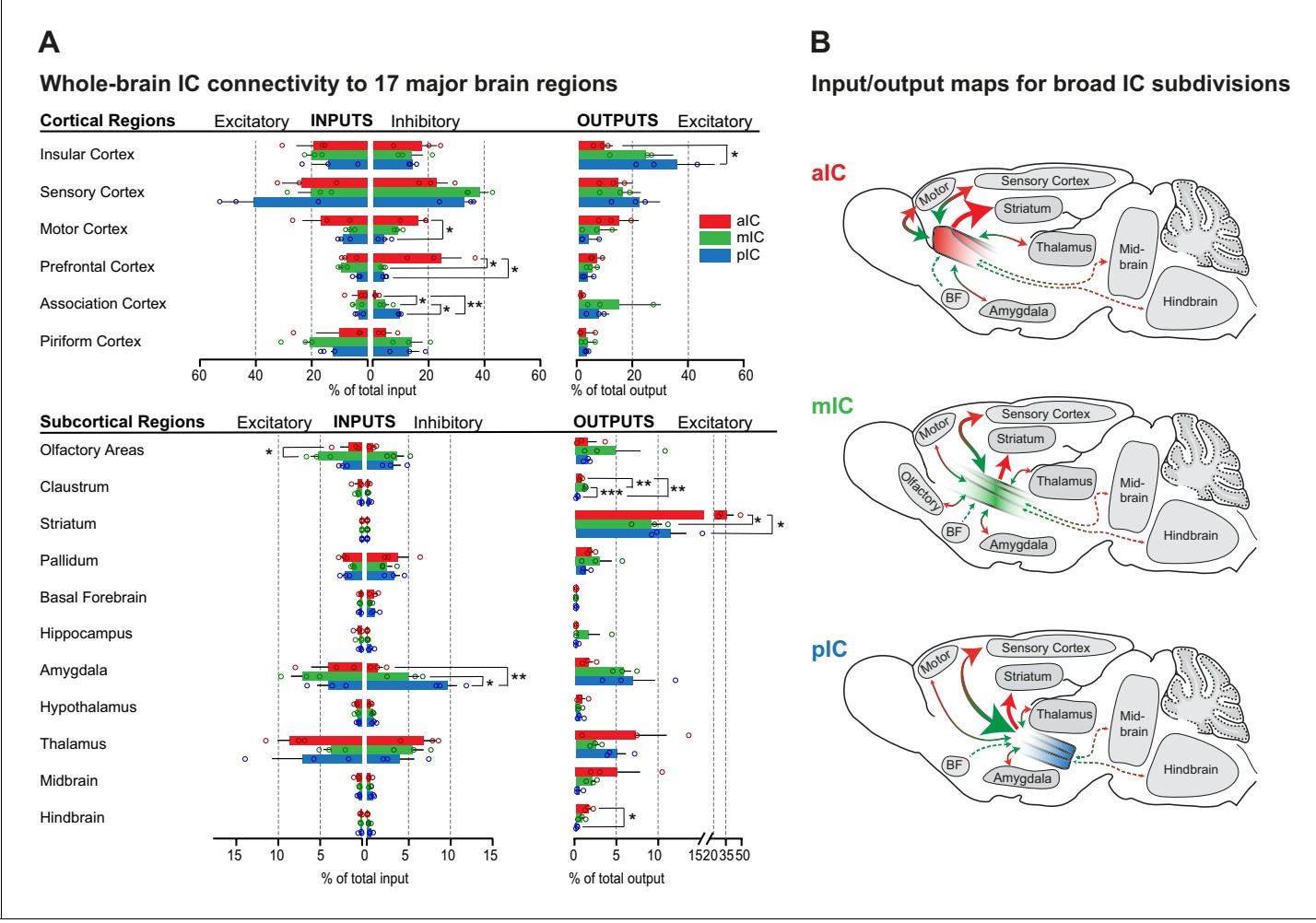

**Figure 2.** Whole-brain IC connectivity map. (**A**) Comparison of inputs to excitatory and inhibitory IC neurons (left) and outputs of excitatory neurons of the IC (right) of all three IC subregions (aIC, red; mIC, green; pIC, blue) across the 17 major brain regions that displayed connectivity. Region values are given as percentage of total cells (RV) or of total pixels (AAV). Data is shown as average ± SEM. n = 3 mice per condition. Top panel shows cortical connectivity, bottom panel shows subcortical connectivity. One-way ANOVAs per subregion followed by Tuckey's multiple comparison test were performed to generate p-values. Significant differences between inputs to excitatory or inhibitory neurons to IC subregions or between outputs from the IC subregions were labeled as ***p<0.001, **p<0.01, *p<0.05. For detailed statistics see ***Supplementary file 3***. (**B**) Individual input-output maps for the three IC subdivisons highlighting selected brain regions. Weight of arrowhead and thickness of arrow shaft indicate strength of connection. Green arrowheads indicate inputs, red arrowheads indicate outputs.

The online version of this article includes the following figure supplement(s) for figure 2:

**Figure supplement 1.** Brain-wide dataset for aIC.
**Figure supplement 2.** Brain-wide dataset for mIC.
**Figure supplement 3.** Brain-wide dataset for pIC.
**Figure supplement 4.** Instructions to query the datasets with custom questions.

thalamus, we describe the IC connectivity to these major interactions partners in more detail in the following sections.

## IC-amygdala connectivity

It has been well established that IC and amygdala are heavily interconnected (***Allen et al., 1991***; ***Augustine, 1996***; ***McDonald et al., 1999***; ***Santiago and Shammah-Lagnado, 2005***) and many important brain functions, for example in valence processing or emotion regulation and awareness, have been suggested to rely on this anatomical link. However, we still lack a detailed understanding of the functional interplay of IC and amygdala, a network affected across many psychiatric disorders.

Recent studies in mice have begun to expose functionally distinct projection pathways between the IC and amygdala (*Gehrlach et al., 2019*; *Lavi et al., 2018*; *Schiff et al., 2018*; *Wang et al., 2018*). We thus next analyzed the detailed connectivity between the nuclei of the mouse amygdala and the IC.

As expected, the afferent connectivity from the amygdala to all three IC subregions was provided by cortex-like subregions of the amygdala, including the basolateral amygdala (BLA), the amygdalo-piriform transition area (APir), the cortical amygdala (ACo) and the extended amygdala (EA), but not from striatum-like nuclei such as the central nucleus of the amygdala (CeA) and medial amygdaloid nucleus (MeA) (*Figure 3A–C*).

Interestingly, the APir sent very strong inputs to the IC. Inhibitory neurons of the pIC received very strong APir inputs which differed significantly between IC subregions ($F_{(2,6)}$ = 10.55, *p=0.0109). Posthoc tests showed that pIC received the strongest inputs (pIC vs. mIC *p=0.0469 and pIC vs. aIC *p=0.0121). On the other hand, excitatory neurons of all IC subregions received comparable amounts of APir inputs. Significant differences were also observed for inhibitory inputs from the EA ($F_{(2,6)}$ = 7.297, *p=0.0247), the ACo ($F_{(2,6)}$ = 14.06, *p=0.0054) and the ventral BLA (vBLA) ($F_{(2,6)}$ = 22.80, *p=0.0016). In those regions, a similar pattern as for APir, namely inhibitory neurons of the pIC being more targeted than mIC or aIC, was shown by Tuckey's multiple compari-son test. From the EA, pIC received significantly more inputs than mIC (*p=0.0321), from the ACo, pIC received significantly more inputs than the aIC (**p=0.0055) and from the ventral BLA (vBLA), pIC received more inputs than mIC (*p=0.0244) and aIC (**p=0.0013). Additionally, mIC received more inputs than the aIC from vBLA (*p=0.0494). Concerning input strength to excitatory cells, there were significant differences found between the IC subregions for inputs from the anterior ($F_{(2,6)}$ = 6.009, *p=0.0369) and posterior part ($F_{(2,6)}$ = 43.25, ***p=0.0003) of the BLA (aBLA, pBLA) as well as from the lateral amygdala (LA) ($F_{(2,6)}$ = 7.609, *p=0.0226). In all three cases posthoc tests showed that mIC received more excitatory inputs than aIC and/or pIC. From the aBLA inputs were signifi-cantly more pronounced to the mIC than the pIC (*p=0.0315), from the pBLA significantly more to the mIC than both the pIC (***p=0.0005) and the aIC (***p=0.0004) and from the LA more to the mIC than the pIC (*p=0.0199).

Concerning the outputs emerging from the IC, we found that for the majority of amygdala subnu-clei, the inputs from the IC emerged in a gradient manner with most inputs provided by the pIC, fewer inputs from the mIC and almost no inputs from the aIC. The only exceptions for this trend were the LA, the aBLA, and the EA. (*Figure 3C,D*). Ordinary one-way ANOVAs revealed significant differences among output strength to the aBMA ($F_{(2,6)}$ = 5.803, *p=0.0396) and APir ($F_{(2,6)}$ = 11.37, *p=0.0091). Posthoc Tuckey's multiple comparison tests showed that to the APir, outputs were sig-nificantly stronger from the pIC than aIC (**p=0.0086) or mIC (*p=0.0365). On the other hand, aBMA received more inputs from the mIC than from the aIC (*p=0.0379).

## IC-striatum connectivity

The striatum, the main input region of the basal ganglia, is implicated in optimizing behavior through refining action selection, reward- and aversion processing, habit formation and modulating motor responses (*Graybiel and Grafton, 2015*). Previous work in rodents describing projections to the stri-atum indicated that the IC targeted the ventral and ventro-lateral striatum, converging with projec-tions from piriform cortex (Pir), medial prefrontal cortex (mPFC), perirhinal cortex (PERI) and the BLA (*Hintiryan et al., 2016*; *Hunnicutt et al., 2016*).

We analyzed the detailed connectivity between the IC and the striatum (*Figure 4*), focusing on the IC-to-striatum outputs, given that there was, as expected, no afferent connection from the stria-tum to any IC subregion (*Figure 2A*). Consistent with a previous study (*Hunnicutt et al., 2016*), we found that the ventral regions of the striatum were more innervated by IC projections than dorsal regions (*Figure 4B*). However, the vast majority of the innervations we detected came specifically from the aIC, which displayed both broad and very dense projections across the ventro-lateral cau-date putamen (CPu), spanning almost the entire structure along its rostro-caudal axis (*Figure 4B–D*). A one-way ANOVA further showed that the difference in output strength between IC subregions to the CPu was significantly different ($F_{(2,6)}$ = 25.92, **p=0.0011). Posthoc testing revealed that the sig-nificant difference was due to higher output from the aIC than the mIC (**p=0.0022) and the pIC (**p=0.0017). Furthermore, the nucleus accumbens core (NAcC) and the interstitial nucleus of the posterior limb of the anterior commissure (IPAC) were densely innervated by aIC projections, despite

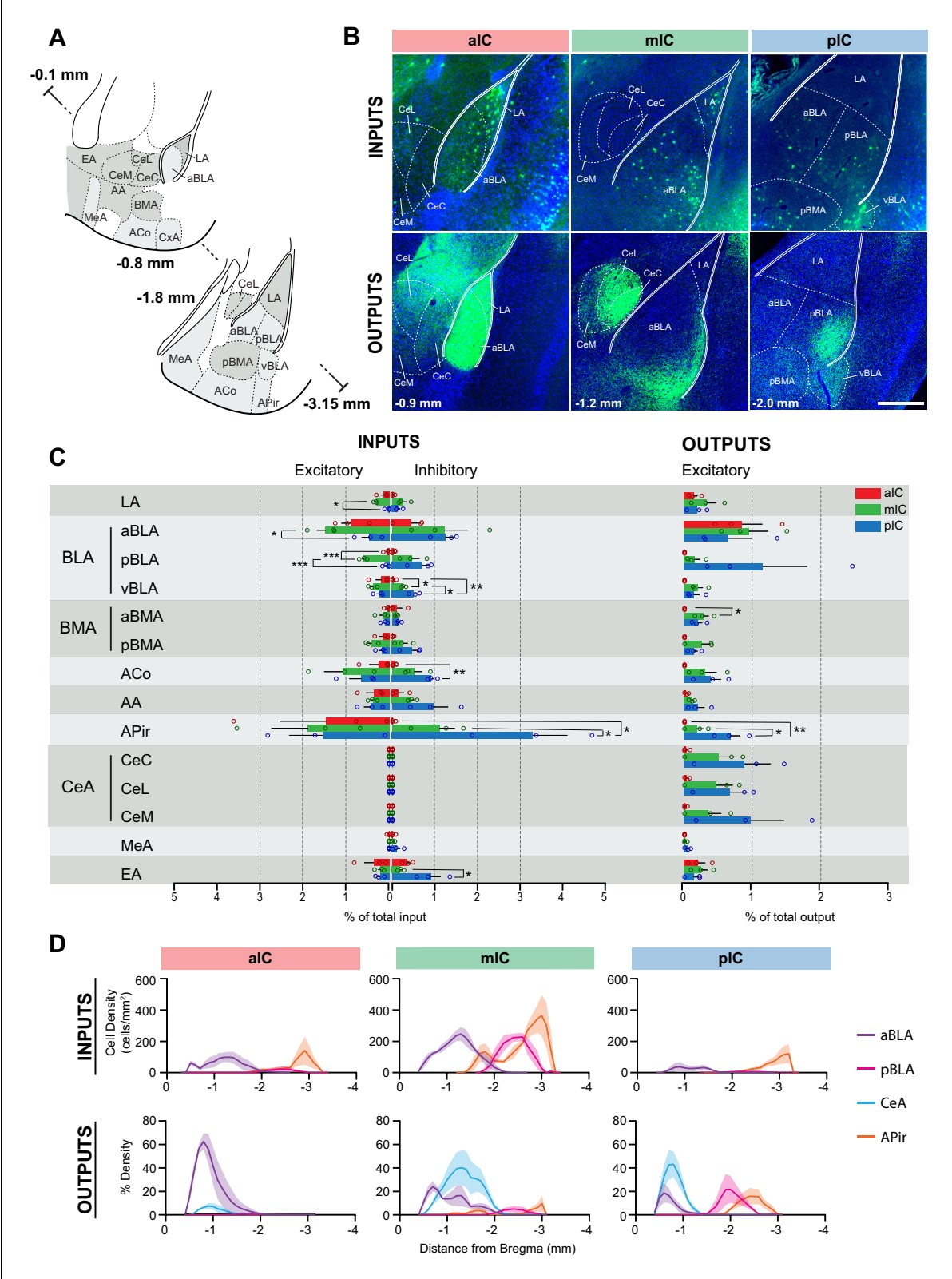

**Figure 3.** IC-amygdala connectivity. (A) Coronal sections depicting the amygdala with its subregions. Distances are provided as anterior-posterior positions relative to Bregma. (B) Representative images from excitatory inputs (top row, eGFP-expressing neurons) and outputs (bottom row, eYFP-positive neurons). Different Bregma levels are shown for each IC target site, as indicated on the images (−0.9 mm, −1.2 mm, −2.0 mm). Scale bar = 200 µm. (C) Comparison of excitatory and inhibitory inputs detected in the amygdala (left) and excitatory outputs from the IC to the amygdala (right) in

*Figure 3 continued on next page*

Figure 3 continued

percent of total in- or output, respectively (aIC, red; mIC, green; pIC, blue). Data is shown as average ± SEM. n = 3 mice per condition. One-way ANOVAs per subregion followed by Tuckey's multiple comparison test were performed to generate p-values. Significant differences between inputs to excitatory or inhibitory neurons to IC subregions or between outputs from the IC subregions were labeled as ***p<0.001, **p<0.01, *p<0.05. For detailed statistics see *Supplementary file 3*. (D) Input cell density (top row) and percent output density (bottom row) plots along the anterior-posterior axis covering the entire amygdala. We selected aBLA, pBLA, CeA and APir to provide the areas with most differences between the IC subregions. n = 3 mice per condition. Data shown as average ± SEM.

their low relative percentage of outputs (*Figure 4B–D*). However, the significant difference between output strength to the IPAC, that was revealed by an one-way ANOVA ($F_{(2,6)}$ = 7.394, **p=0.0240), was due to differences between the pIC and the aIC (*p=0.0312) and between the pIC and the mIC (*p=0.0433).

The mIC and pIC also projected to the CPu, but to a much weaker extent than aIC (approximately 5-fold lower). However, both mIC and pIC densely innervated the IPAC (to around 60% density) (*Figure 4D*). Overall, mIC and pIC showed a very similar connectivity pattern to the striatum with 9% and 11% of total output, respectively. In contrast, aIC output to the striatum represents the largest

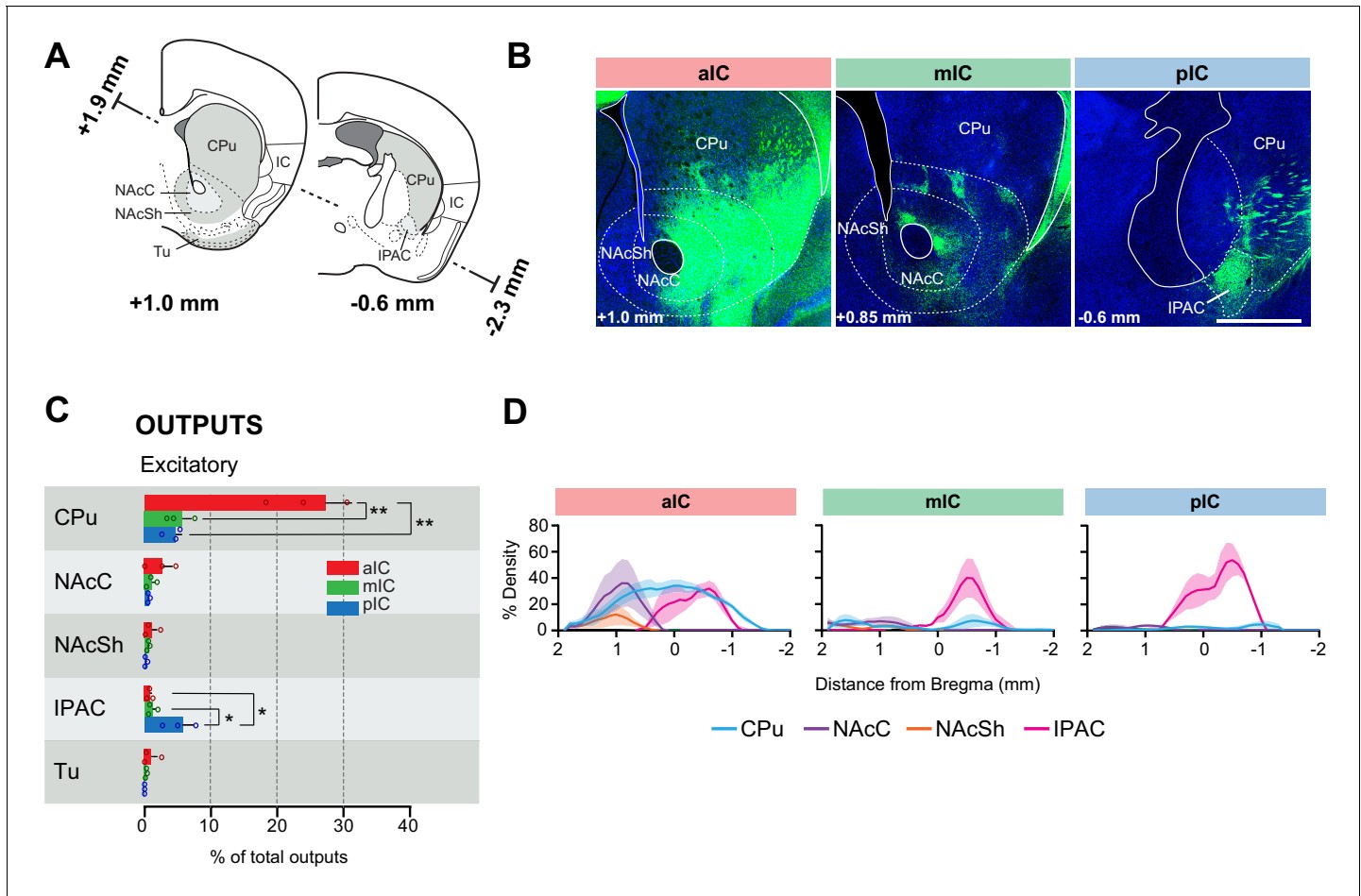

**Figure 4.** IC-striatum connectivity. (A) Coronal sections depicting the striatum with its subregions. (B) Representative images from excitatory outputs (eYFP-positive neurons). Note the dense innervation of CPu, NAcC and NAcSh by the aIC. Different Bregma levels are shown for each aIC, mIC and pIC, as indicated on the images. Scale bar = 500 µm. (C) Comparison of excitatory outputs from the three IC subregions to the striatum in percent of total output (aIC, red; mIC, green; pIC, blue). Values are given as percentage of total pixels. Data shown as average ± SEM, n = 3 mice per condition. One-way ANOVAs per subregion followed by Tuckey's multiple comparison test were performed to generate p-values. For detailed statistics see *Supplementary file 3*. (D) Plots depict the density of IC innervation along the anterior-posterior axis of the striatum. n = 3 mice per condition, data shown as average ± SEM.

output out of all regions innervated by aIC (31.8%). Both, mIC and pIC specifically innervated striatal patches (*Brimblecombe and Cragg, 2017*), as seen for mIC in *Figure 4B*.

Taken together, we found a large difference in the innervation of the striatum along the rostro-caudal axis of the insula, with the aIC providing the strongest projections.

## IC-thalamic connectivity

We next assessed the third largest subcortical connectivity partner of the IC: the thalamus. Thalamo-cortical projections are thought to be essential drivers of cortical activity in sensory areas and associative brain regions (*Hunnicutt et al., 2014*). Cortico-thalamic feedback projections stemming from layer 6, in turn, shape thalamic cell activity via monosynaptic and disynaptic connections (*Crandall et al., 2015*). The function of cortical regions has often been inferred by characterizing the type of thalamic input they receive (*Sherman and Guillery, 2006*).

The afferent connectivity to the aIC originated mainly from higher-order associative and motor nuclei, with the majority of inputs arising from the polymodal association group of thalamic nuclei (medio-dorsal (MD) and centro-median (CM) nuclei) (*Figure 5*). Furthermore, the aIC and mIC received innervation from two sensory-motor related nuclei, the ventro-medial (VM) and the ventral anterio-lateral (VAL) nucleus. Input strength from the CM, as revealed by one-way ANOVAs, was significantly different between IC subregions to both excitatory ($F_{(2,6)}$ = 15.61, **p=0.0042) and inhibitory neurons ($F_{(2,6)}$ = 21.56, **p=0.0018). In both cases, posthoc testing showed that this difference was due to stronger inputs to the aIC compared with both other IC subregions for both the excitatory (aIC vs. mIC **p=0.0082, aIC vs. pIC **p=0.0059) and the inhibitory inputs (aIC vs. mIC and aIC vs. pIC **p=0.0031) (*Figure 5C*). The afferents of pIC, on the other hand, originated majorly from sensory-related nuclei, with the greatest inputs from the posterior complex (Po) and the ventral posterior complex (VPC). While the pIC also received inputs from the MD, it was only weakly innervated by the CM. A one-way ANOVA revealed a significant difference in medial geniculate nucleus (MGN) inputs to excitatory neurons between different IC subregions ($F_{(2,6)}$ = 12.23, **p=0.0076). Tuckey's multiple comparison tests showed that inputs were stronger to the pIC compared with both aIC (*p=0.0119) and mIC (*p=0.0126). Interestingly, the afferents of mIC exhibited characteristics of both aIC and pIC, receiving projections from sensory-, motor- related, and higher order thalamic nuclei.

As expected from thalamo-cortical pathways (*Hunnicutt et al., 2014*), IC outputs reciprocated their thalamic inputs. For example, the aIC strongly and densely innervated the VM, MD and CM, thus putatively closing the thalamo-cortico-thalamic loop. The strongest aIC projection innervated the VM, and these projections tended to be stronger than projections from the mIC or pIC (*Figure 5C*, right). The pIC strongly and densely innervated the VPC in particular, and had almost no projections to any other thalamic nucleus. Furthermore, there was a significant difference found by one-way ANOVA between output strengths to the VP ($F_{(2,6)}$ = 9.316, *p=0.0145). Overall, the VP received its strongest inputs from the pIC and much less input from aIC (*p=0.0224) or mIC (*p=0.0260) (*Figure 5C*, right).

## Bidirectional connectivity

We next investigated the reciprocity of the IC connectivity with other brain areas by correlating inputs to excitatory neurons with their respective outputs (*Figure 6A*).

We first assessed the reciprocity of the connections between the IC and other cortical regions. We found a significant correlation for the connectivity of the mIC and pIC with other cortical regions and a strong trend for correlation for the mIC with other cortical regions. Thus, the IC was mostly bidirectionally connected to many other cortical regions.

Subcortical regions, on the other hand, were most often not bidirectionally connected to the IC. Instead, we could define many connections with the subcortex as either being IC input-dominated (stronger subcortical-to-cortical connectivity, *Figure 6B*, green) or IC output-dominated (stronger cortical-to-subcortical connectivity, *Figure 6B*, blue). No region strongly reversed its connectivity characteristic when comparing between aIC, mIC and pIC. However, the mid- and hindbrain nuclei received less pIC innervation compared to aIC and mIC (e.g. compare raphe nuclei (RN) input vs output coordinates in *Figure 6A*), suggesting that pIC has a less direct influence on neuromodulatory systems.

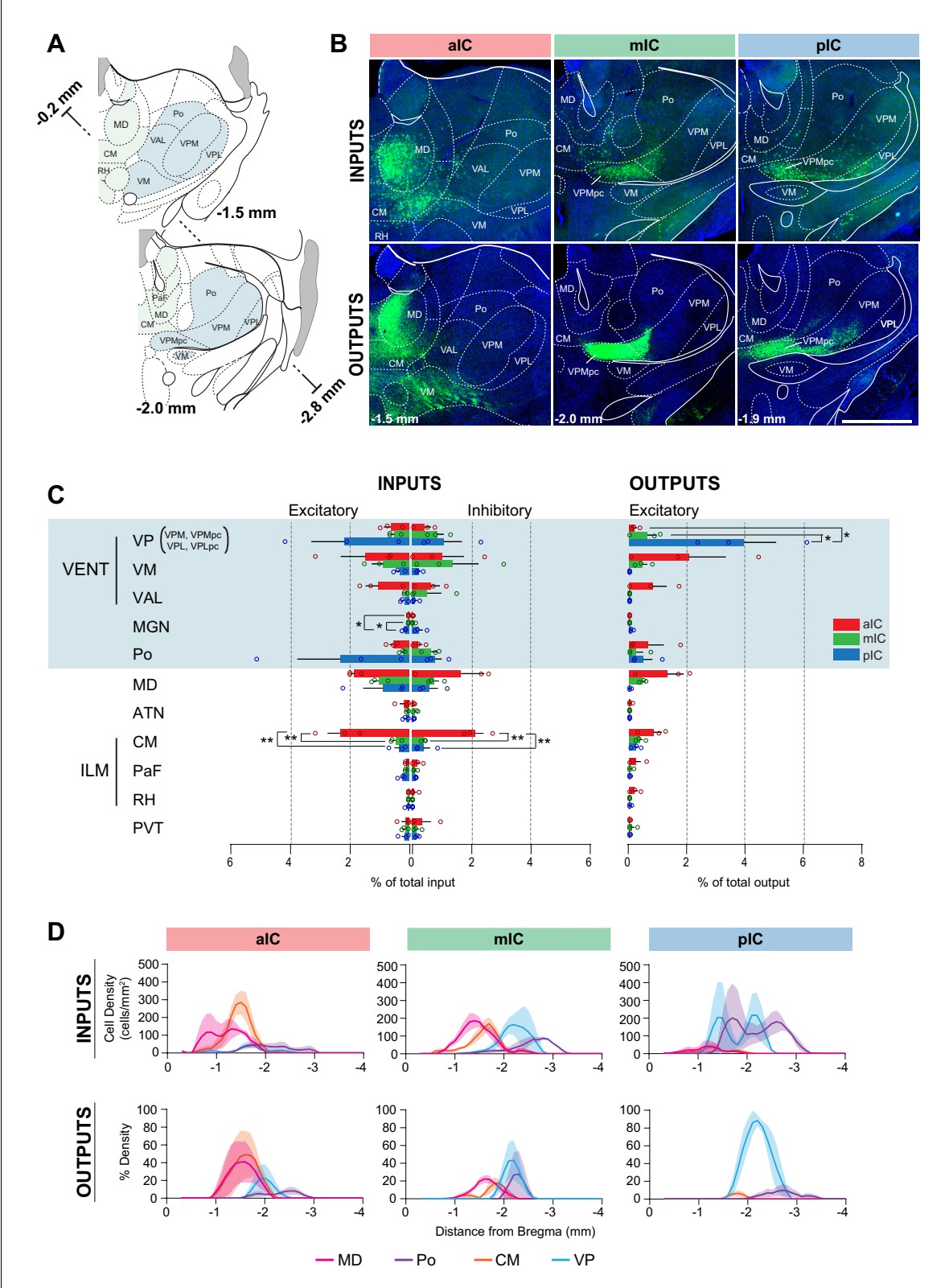

**Figure 5.** IC-thalamus connectivity. (**A**) Coronal sections depicting the thalamus with subregions that connect to the IC. (**B**) Representative images from excitatory inputs (top row, eGFP-expressing cell bodies) and outputs (bottom row, eYFP-positive neurons). Different Bregma levels are shown for each IC subregion as indicated on the images. Scale bar = 500 μm. (**C**) Comparison of inputs to excitatory or inhibitory neurons of all three IC subregions (left) and of outputs from excitatory IC neurons to the thalamus (aIC, red; mIC, green; pIC, blue). Values are calculated as percentage of total cells (RV)

*Figure 5 continued on next page*

*Figure 5 continued*

or of total pixels (AAV). Data shown as average ± SEM, n = 3 mice per condition. One-way ANOVAs per subregion followed by Tuckey's multiple comparison test were performed to generate p-values. Significant differences between inputs to excitatory or inhibitory neurons to IC subregions or between outputs from the IC subregions were labeled as **p<0.01, *p<0.05. For detailed statistics see *Supplementary file 3*. (D) Input cell density (top row) and output density (bottom row) plots along the anterior-posterior axis. Thalamus regions of interest are shown, n = 3 mice per condition, data shown as average ± SEM.

This analysis revealed, that amongst the amygdala subnuclei, the CeA was strongly innervated by the IC without sending backward projections. The opposite was true for subnuclei such as the APi, AA, ACo, or BMA, who mostly send projections to the IC or were bidirectionally connected. Interestingly, the thalamus was mostly bidirectionally connected with the IC, whereas the striatum, midbrain and hindbrain connectivity was mostly dominated by projections from the IC to the subcortical regions (*Figure 6B*, bottom).

However, this analysis did not address reciprocity of connectivity at the level of single neurons but rather brain regions. Thus, inhibitory neurons in the IC could receive inputs from a brain region that is densely innervated by IC axons. For this reason, we also compared reciprocity of connections for inhibitory neurons (*Figure 6—figure supplement 1*). In line with the finding that we detected very few overall differences between excitatory and inhibitory cell connectivity, this analysis revealed very similar results as shown in *Figure 6A* for excitatory neurons.

## Comparison of input and output distributions

Throughout our analyses, we have seen distinctions arising between the three IC subregions we targeted. To test whether these observations represent meaningful differences, we correlated in an unbiased manner all input tracings to each other (including inhibitory and excitatory connectivity experiments). We additionally performed the same analysis for all output tracings. We compared the 17 major brain regions in a pairwise fashion and hierarchically clustered the correlation coefficients (*Figure 7* and Materials and methods). Overall, there was a high degree of similarity for the input-input comparisons (average correlation coefficients of 0.7 ± 0.16), and, to a lesser extent, for the output-output comparison (average correlation coefficients of 0.45 ± 0.28). However, for both inputs and outputs, two distinct clusters did form, separating the aIC tracings from a grouped mIC/pIC pool. Furthermore, for both inputs and output correlations (*Figure 7A,B*), the mIC and pIC tracings were so similar that they did not fall into separate clusters. Indeed, the relative location of the starter cell population (left columns, green gradient) did not lead to a separate clustering of mIC and pIC targeted tracings. Finally, for the input data, there was no correlation separating excitatory and inhibitory tracings, supporting our conclusions stated earlier that the IC afferents for these two cell types is similar.

Taken together, the input- and output-patterns of aIC suggest a functional difference compared to mIC and pIC regions. In particular, for the output network, a key difference arises from the high degree of efferent connectivity between the aIC and striatum, and to a lesser extent, the motor cortex, as compared with the mIC and pIC (*Figure 2A*). For inputs to the IC, the difference is not so profound, with subtle variations in regions such as the motor cortex (as with outputs, biased towards the aIC) the sensory cortex (pIC-biased), the amygdala (slight pIC/mIC bias) and producing the two clusters.

## Discussion

In this study, we systematically mapped the brain-wide input- and output connectivity of inhibitory and excitatory neurons of three subregions of the mouse insular cortex. All IC subregions exhibit multifaceted and brain-wide connectivity patterns, with a substantial degree of intra-insular cross talk. These factors result in a series of complex, multi-modal hubs, suggesting that each subregion is not limited to a single specialized function.

By performing unbiased cluster analysis, we found differences in IC connectivity along the rostral-caudal axis, in particular in regards to both in- and outputs to and from the aIC as compared to those of the mIC and pIC. For the outputs, this difference is in part due to a specifically strong ventro-lateral striatum innervation by the aIC. Overall, the aIC also showed a bias toward connectivity

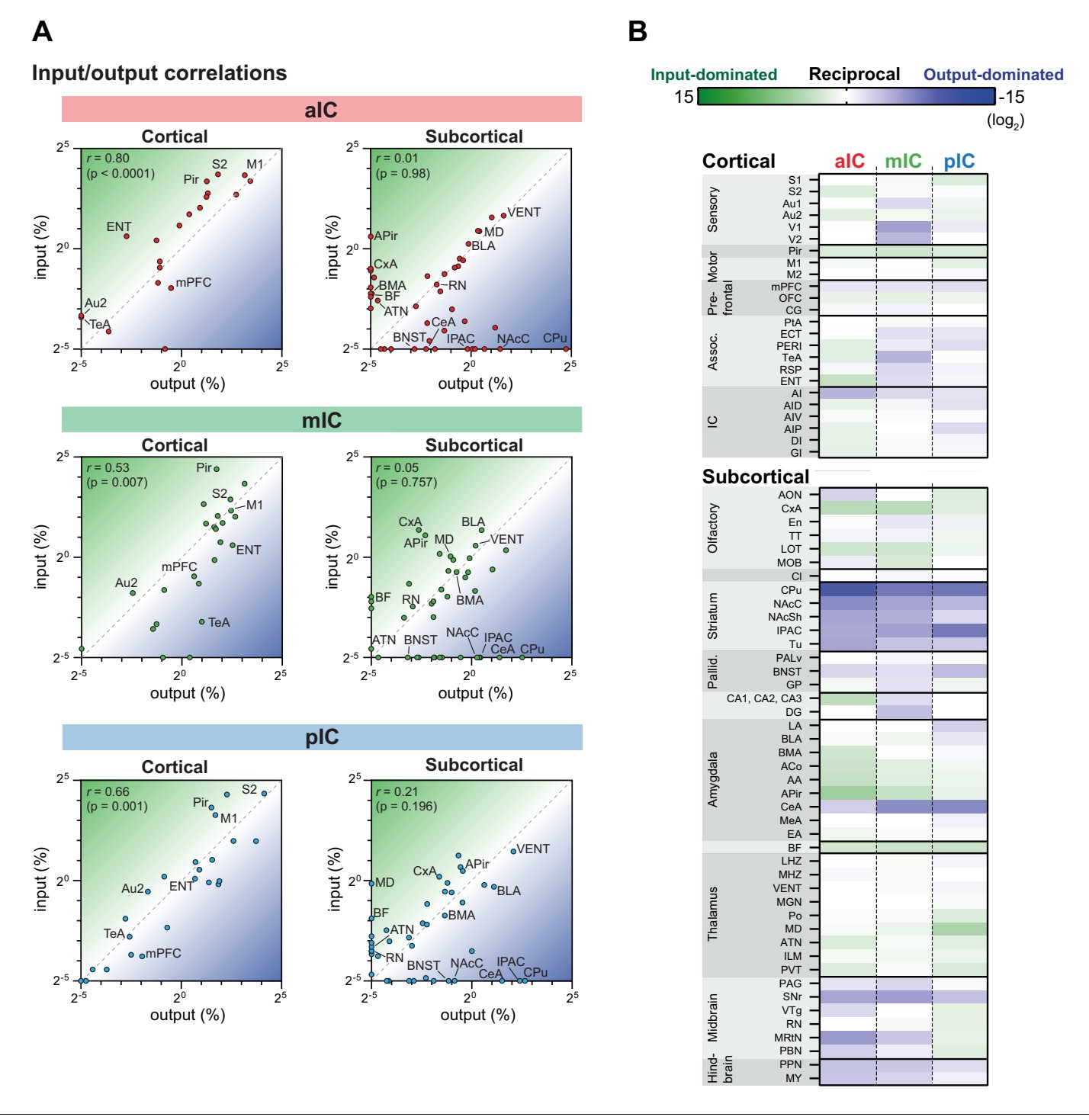

**Figure 6.** IC input-output relationships for excitatory cells. The global dataset was further subdivided into subregions of higher specificity (see *Figure 2—figure supplements 1–3*). (A) The average value for each excitatory input and output was correlated for the three IC subregions. Data is divided into cortical (left panels) and subcortical (right panels) regions. Subregions that lacked both input and output neurons are not included in the graphs. Note the high correlation in the cortical connectivity as compared to the connectivity in the subcortex for all datasets (*r* = Pearson's correlation coefficients). (B) Heatmaps showing fold-difference between inputs to outputs per brain subregion for each IC target. Green gradient represents connectivity characterized by stronger inputs to the IC from target regions, blue gradient represents connectivity characterized by stronger projections from the IC to target regions. Subregions where no signal was detected for both input and output conditions were omitted. Data shown as ratio from the average of three mice per condition per IC subregion. The meaning of the abbreviations can be found in *Supplementary file 1*.

*Figure 6 continued on next page*

*Figure 6 continued*

The online version of this article includes the following figure supplement(s) for figure 6:

**Figure supplement 1.** Input-output relationships for excitatory output and inhibitory input.

with locomotion-related areas, such as the motor cortices (M1, and M2), the ventro-medial (VM) and centro-median thalamic (CM) nuclei and, the substantia nigra and the midbrain reticular nucleus. This connectivity pattern may provide an anatomical foundation for why optogenetic stimulation of the aIC can elicit appetitive and seeking behavior and has been described as a 'positive valence' region (*Peng et al., 2015*; *Wang et al., 2018*). Overall, our data suggest a general role of the aIC with functional roles beyond that of a 'sweet cortical field' (*Peng et al., 2015*), since intra-insular projections from the mIC and pIC, which process diverse bodily information, are one of its main input sources. Based on its connectivity and knowledge gained in previous functional studies, the aIC could serve as an integrator of positive-valence signals that then guide motivated behavior through its downstream projections, in particular via the ventral striatum and motor cortex. Given our approach we could not dissect further differences between the dorsal (AID) and ventral division of aIC (AIV), but studies performed in hamsters and rats have suggested a further distinction of projection patterns between the AID and AIV (*Hintiryan et al., 2016*; *Maffei et al., 2012*; *McDonald et al., 1999*; *Reep and Winans, 1982*).

Interestingly, our correlation and clustering analysis suggests that the mIC is more similar to the pIC. The GI/DI of the mIC is referred to as the 'gustatory cortex' in the Allen Mouse Brain Atlas (*Lein et al., 2007*). Notably, we found that the mIC received strong olfactory related inputs (from

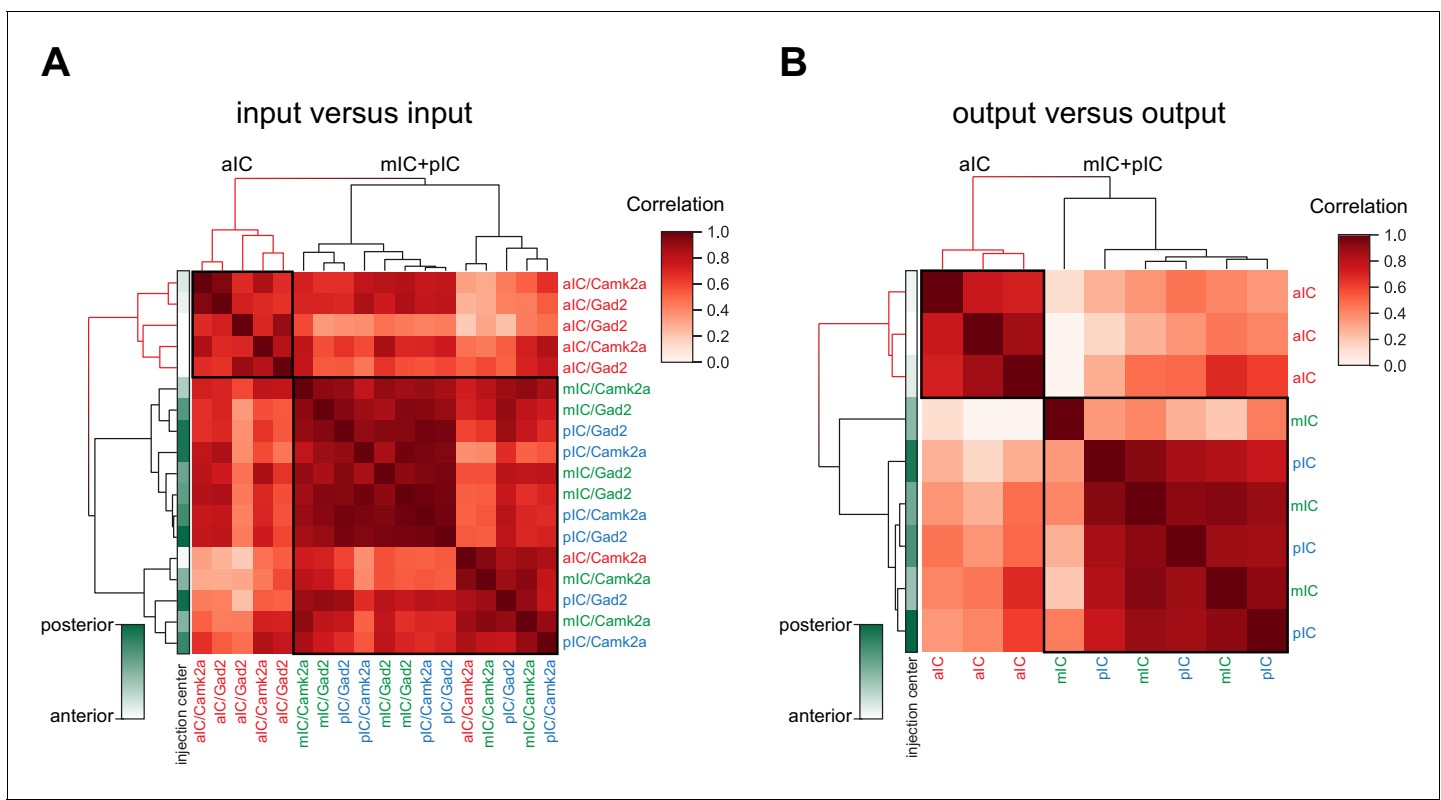

**Figure 7.** Connectivity-based subregions of the IC. Matrices of hierarchically clustered pair-wise correlation coefficients (Pearson's) of animals (A) inputs vs. inputs (N = 18 mice) or (B) outputs vs outputs (N = 9 mice). The pair-wise correlations were performed on the data organized into 17 major brain regions (see *Figure 2*). Far left gradient bar (green hues) indicates the center of the starter cells, ranked relative to every mouse in the dataset. Note for both input and output correlations, a clear cluster forms from the aIC-targeted animals (top left boxed sections and red-colored dendrograms), whereas the mIC- and pIC-targeted animals intermingle in a second cluster (larger boxed areas). Interestingly, the clustering algorithm did not separate excitatory (*Camk2a*-Cre) from inhibitory (*Gad2*-Cre) rabies virus tracings.

the Pir, APir, and piriform-amygdalar area (CxA)). Furthermore, compared to the aIC and pIC, the mIC sends more outputs to memory related areas, such as the entorhinal cortex and the ventral hippocampus. Indeed, previous studies demonstrated the role of the mIC in conditioned taste aversion (CTA) (*Lavi et al., 2018*). Taken together with previous functional studies, our anatomical description supports a role for the mIC as a learning hub, involved in various aspects of consummatory behaviors, such as texture processing of food, palatability, taste aversion or preference. Potentially, this could extend past consummatory domains, into social transmission of food preferences or reproductive behavior.

We also extended our recent findings of pIC connectivity (*Gehrlach et al., 2019*), and can now clearly establish it as an anatomically defined subregion of the mouse IC through direct comparison of its connectivity as compared to the aIC, in particular. The pIC exhibits a multimodal convergence of inputs from subcortical sites carrying bodily and limbic information streams. In addition, it sends strong projections to subcortical regions implicated in emotional and motivated functions (*Simmons et al., 2013*). This includes innervation of sensory, autonomic, motor associative and limbic structures. Furthermore, there are more intra-insular outputs from the pIC than from the aIC, implying a caudal-to-rostral flow of information, as has been suggested previously (*Craig, 2009*; *Fujita et al., 2010*).

Our analysis of the reciprocity of IC connectivity with other cortical or subcortical regions revealed strong correlations between in- and outputs for cortical regions across all IC subregions. In contrast, many subcortical regions sent strong projections to the IC (raphe nuclei (RN), basal forebrain (BF), olfactory regions, and thalamic nuclei) or predominantly received axonal inputs from the IC (CeA, striatum, SNr, and BNST), revealing strong directionality of connections between the IC and subcortical regions. These comparisons come with the caveat that we have no physiological measurements of relative connectivity strength.

Interestingly, inhibitory interneurons, irrespective of which IC subregion was analyzed (APir aside), displayed very similar connectivity patterns and strength when compared to excitatory pyramidal neurons. This was supported by our correlation and hierarchical cluster analysis and is in agreement with several rabies virus tracings studies in other brain regions (*Beier et al., 2015*; *Do et al., 2016*; *Luo et al., 2019*; *Wall et al., 2016*) and may underlie the balance of excitation and inhibition in cortex (*Sohal and Rubenstein, 2019*; *Yizhar et al., 2011*).

Although this investigation sought to systematically compare brain-wide IC connectivity, there are limitations that need to be considered.

First, despite there being good separation between the bulk of the starter cell populations into our defined IC subregions, there is a small proportion of overlap that may influence the connectivity. This only affects the results when comparing with mIC results, as aIC and pIC starter populations were completely separated. The exact extent of starter cell distributions into other subregions is visualized in *Figure 1C* and *Figure 1—figure supplement 2C*.

Contamination into ventrally or dorsally neighboring non-IC regions may also influence the results we observed. We thus quantified the extent of starter-cell spillover into neighboring regions (*Figure 1—figure supplement 2C*).

First, we considered whether there was a correlation between the amount of spillover in each individual brain and the overall strength of connectivity between the IC and major brain regions. We did not find that tracings with more spillover would systematically exhibit more connectivity to other regions of the brain compared to the same tracing with less spillover, confirming that the amount of spillover did not qualitatively bias our results (*Figure 1—figure supplement 2D*). However, while spillover was very limited for most tracing conditions, there was a sizable amount of spillover into Pir specifically for the input tracings to the excitatory neurons of the aIC. We thus considered in detail whether this spillover may explain the differences we report for aIC versus mIC and pIC connectivity.

In *Figure 2A*, we detected one significant difference for inputs onto excitatory neurons between the IC subregions. This difference was a significantly stronger innervation from olfactory areas to the mIC than the aIC. Because the inputs were stronger onto mIC these differences cannot be explained by the piriform contamination of aIC starter cells. All other significant differences detected between major brain regions and the IC subregions (*Figure 2*) were found for inputs onto inhibitory neurons, which did not suffer from contamination.

We next considered what is known about Pir connectivity. The Pir is known to receive strong inputs from olfactory areas (*Wang et al., 2020*). Therefore, if our results reported for excitatory input tracings to the aIC were largely affected from the starter cell contamination in the Pir, we would expect to have significantly higher innervation of the aIC compared to both the mIC and the pIC. However, this is not what we observe arguing that the Pir contamination does not drive a major qualitative difference in our tracings from aIC.

Taken together, while it is important to note that our aIC rabies tracings onto excitatory neurons contain a sizeable amount of Pir starter cells, based on the nature of differences we observed between aIC and mIC/pIC connectivity as well as existing Pir tracing results, we do not believe that this spillover affects the main conclusions on aIC versus mIC/pIC connectivity we make in this study (*Figure 1—figure supplement 2C–D*).

In order to limit spillover into the neighboring motor and somatosensory cortices, injection sites were aimed toward the center of the IC. As a consequence, the amount of starter cells in the GI was overall smaller than in DI or AI (*Figure 1—figure supplement 2C*). Furthermore, we detected that in AAV tracings, we traced from comparatively more starter cells in the AI in the aIC compared to mIC/pIC. However, this bias was not present for excitatory rabies tracings. The true starter cell distributions should always be considered when interpreting tracing data.

It is furthermore noteworthy that one major limitation of the datasets presented here is the lack of tracings to and from the brainstem. Previous studies have shown strong connections between IC and brainstem nuclei such as NTS and DMV (*Cobos et al., 2003*; *Gaytán and Pásaro, 1998*; *Saper and Stornetta, 2015*; *Shipley, 1982*). While we did not address this connectivity due to practical reasons, future studies will be required to comprehensively map the connectivity between the IC and the brainstem in the mouse.

As AAV tracing originating from three different starter locations and differing sizes of starter cell populations led to varying degrees of axonal labeling, acquisition settings had to be adjusted for each brain individually. In order to be able to interpret results, we therefore display data as percentage of total. This type of presentation comes with the caveat that smaller regions are underrepresented, for example the DRN that only provided below 0.1% of the total inputs consistently projected to the IC. Thus functional implications cannot solely be determined from the relative number of inputs. The alternative, using density analysis, underrepresents larger areas, such as the cortical regions (please refer to the pivot table in the *Supplementary file 2* which allows custom plots of densities for all data presented here).

Additionally, counting labeled fiber presence after AAV infection to detect the output strength does not directly represent synaptic connectivity. Recent technology using a fluorescent protein tagged to a synaptic marker (e.g. AAV-DIO-mRuby-T2A-synaptophysin-eGFP, [*Knowland et al., 2017*]) or the trans-synaptic infection of AAV1-Cre (*Zingg et al., 2017*) would overcome this limitation.

Lastly, even though retrograde tracing with G deleted rabies virus is a powerful tool for circuit tracings, it still comes with technical limitations. As detailed by *Callaway and Luo, 2015*, the labeling of inputs is only partial and could be biased toward certain cell types, differential labeling of active versus inactive synapses, or neuromodulatory versus fast neurotransmitter synapses. More recently, it has also been shown that uptake efficiency and spread of rabies virus is influenced by neuronal activity (*Beier et al., 2017*). Given these biases in transsynaptic spread and in the absence of physiological measurements of synaptic strength, the anatomical findings reported here should be regarded as rather qualitative since we cannot draw conclusions about the strength of a given connection.

Bearing this in mind, this study highlights specific IC connectivity patterns that warrant further functional investigation. These include the localized targeting of pIC projections to the IPAC, which may be involved in motivated behaviors like approach, seeking and feeding. In the amygdala, all IC regions project to the BLA, but only the pIC innervated the pBLA. Understanding the role of this pathway would help in both describing IC function and the specificity of amygdala subregions. The APir is also preferentially targeted by pIC projections, and reciprocates this connection, which may provide an interesting pathway for odor-related responses.

Accompanying this study, we provide an excel sheet that contains the entire dataset (*Supplementary file 2*). Using the pivot table function of Microsoft Excel allows to recreate any plot presented in this study and to query and reanalyze the datasets for individual questions. In the excel

sheet, we provide five example pivot tables and describe the workflow to create such tables in *Figure 2—figure supplement 4*.

Taken together, our dataset combined with functional studies suggest that the insula is a hub that integrates bodily information with memory and emotional content and to guide behavior and maintain homeostasis.

# Materials and methods

**Key resources table**

| Reagent type (species) or resource | Designation | Source or reference | Identifiers | Additional information |
|---|---|---|---|---|
| Genetic reagent (*M. musculus*) | *Camk2a-Cre* | https://www.jax.org/strain/005359 | IMSR Cat# JAX:005359, RRID:IMSR_JAX:005359 | B6.Cg-Tg(Camk2a-cre) T29-1Stl/J |
| Genetic reagent (*M. musculus*) | *GAD2-Cre* | https://www.jax.org/strain/010802 | IMSR Cat# JAX:010802, RRID:IMSR_JAX:01002 | Gad2tm2(cre)Zjh/J |
| Software, algorithm | CellProfiler 3.0.0 | https://cellprofiler.org/ | CellProfiler Image Analysis Software, RRID:SCR_007358 | |
| Software, algorithm | FIJI | Fiji is just ImageJ, NIH (https://imagej.net/Fiji) | Fiji, RRID:SCR_002285 | |
| Software, algorithm | Autonomous_neuron_detection.ijm | This paper, GitHub (https://github.com/GogollaLab/tracing_quantification_and_analysis/blob/master/autonomous_neuron_detection.ijm) | | |
| Software, algorithm | Counting_RV.ijm | This paper, GitHub (https://github.com/GogollaLab/tracing_quantification_and_analysis/blob/master/counting_RV.ijm) | | |
| Software, algorithm | Roi_set_atlas | This paper, GitHub (https://github.com/GogollaLab/tracing_quantification_and_analysis/tree/master/ROI_set_atlas) | | |
| Software, algorithm | Leica Application Suite X 3.3.0.16799 | https://www.leica-microsystems.com/products/microscope-software/details/product/leica-las-x-ls/ | Leica Application Suite X, RRID:SCR_013673 | |
| Software, algorithm | Autonomous_pixel_detection.ijm | This paper, GitHub (https://github.com/GogollaLab/tracing_quantification_and_analysis/blob/master/autonomous_pixel_detection.ijm) | | |
| Software, algorithm | Counting_AAV.ijm | This paper, GitHub (https://github.com/GogollaLab/tracing_quantification_and_analysis/blob/master/counting_AAV.ijm) | | |
| Software, algorithm | Python 3.6 | http://www.python.org/ | Python Programming Language, RRID:SCR_008394 | |
| Software, algorithm | Analysis_RV.py | This paper, GitHub (https://github.com/GogollaLab/tracing_quantification_and_analysis/blob/master/analysis_RV.py) | | |
| Software, algorithm | Analysis_AAV.py | This paper, GitHub (https://github.com/GogollaLab/tracing_quantification_and_analysis/blob/master/analysis_AAV.py) | | |
| Software, algorithm | GraphPad Prism | GraphPad Software, CA (https://graphpad.com) | GraphPad Prism, RRID:SCR_002798 | |
| Other | AAV2/5-EF1a-DIO-eYFP | UNC Vector Core https://www.med.unc.edu/genetherapy/vectorcore/ | In-Stock AAV Vectors – Dr. Karl Deisseroth, 100 ul Aliquots | $5.6 \times 10^{12}$ vg/ml |

*Continued on next page*

*Continued*

| Reagent type (species) or resource | Designation | Source or reference | Identifiers | Additional information |
|---|---|---|---|---|
| Other | AAV2/8-EF1a-FLEX-TVA-mCherry | UNC Vector Core https://www.med.unc.edu/genetherapy/vectorcore/ | In-Stock AAV Vectors – Dr. Karl Deisseroth, 100 ul Aliquots | $4.2 \times 10^{12}$ vg/ml |
| Other | AAV2/8-CA-FLEX-RG | UNC Vector Core https://www.med.unc.edu/genetherapy/vectorcore/ | In-Stock AAV Vectors – Dr. Karl Deisseroth, 100 ul Aliquots | $2.5 \times 10^{12}$ vg/ml |
| Other | SADΔG-eGFP(EnvA) | UNC Vector Core https://www.med.unc.edu/genetherapy/vectorcore/ | In-Stock AAV Vectors – Dr. Karl Deisseroth, 100 ul Aliquots | $3 \times 10^{8}$ ffu/ml |

## Animals

Mice between 2 and 6 months of age were used in accordance with the regulations from the government of Upper Bavaria (Animal license AZ: 55.2-1-54-2532-56-2014). *Camk2a*-Cre (B6.Cg-Tg (Camk2a-cre)T29-1Stl/J) mice were used for both retrograde rabies virus tracings and anterograde axonal tracings. Retrograde rabies virus tracings were also performed in *Gad2*-Cre (Gad2tm2(cre) Zjh/J) mice. Both female and male mice were employed (Fig S1c). For controls, we used male C57Bl6\NRj mice. All mice group housed 2–4 mice/cage and were kept on an inversed 12 hr light/dark cycle (lights off at 11:00 am). Mice were provided with ad libitum access to standard chow and water.

## Viral constructs

Unless otherwise stated, the following constructs were obtained from the UNC Vector Core (Gene Therapy Center, University of North Carolina at Chapel Hill, USA). For anterograde tracings AAV2/5-EF1α-DIO-eYFP ($5.6 \times 10^{12}$ vg/ml) was used. For retrograde rabies virus tracings AAV2/8-EF1α-FLEX-TVA-mCherry ($4.2 \times 10^{12}$ vg/ml), AAV2/8-CA-FLEX-RG ($2.5 \times 10^{12}$ vg/ml), and G-deleted EnvA-pseudotyped rabies virus -eGFP (SADΔG-eGFP(EnvA)) ($3 \times 10^{8}$ ffu/ml), were prepared as described before (Gehrlach et al., 2019; Wickersham et al., 2007b).

## Surgeries

Anesthesia was initiated with 5% isoflurane and maintained at 1–2.5% throughout surgery. Metamizol (200 mg/kg, s.c., WDT, Garbsen, Germany) was injected for peri-operative analgesia and carprofen (s.c., 5 mg/kg, once daily for 3 days, Zoetis) for post-operative pain management. Mice were secured in a stereotaxic frame (Stoelting, IL), placed on a heating pad (37°C) and eye ointment (Bepanthen, Bayer) was applied. For viral infusions, pulled glass-pipettes were attached to a microliter syringe (5 μL Model 75 RN, Hamilton, NV) using a glass needle compression fitting (#55750–01, Hamilton), mounted on a syringe pump controlled by a microcontroller (UMP3 + micro4, WPI). After trepanation of the skull, mice were unilaterally injected with 100–150 nL of a 6:1 (RG: TVA) mixture of helper-viruses. The following coordinates (mm from Bregma) were used: for anterior IC: AP: +1.9 mm, ML: + or - 2.7 mm, DV: −3.0 mm. For medial IC: AP: 0.7 mm, ML: + or – 3.7 mm, DV: −4.0 mm. For posterior IC: AP: −0.5 mm, ML: + or – 4.05 mm, DV:- 4.0 mm. The trepanation was sealed with bone wax and the skin sutured. After 3–4 weeks, 350 nL of SADΔG-eGFP(EnvA) was injected into the same coordinates. Mice were sacrificed 7 days after infusion of the rabies virus. For axonal AAV-tracings in *Camk2a*-Cre mice, AAV2/5-EF1α-DIO-eYFP (80–100 nl) was injected unilaterally into either the aIC, mIC or pIC coordinates mentioned above. Mice were sacrificed 4 weeks after the injections.

## Histology

Animals were anesthetized with ketamine/xylazine (100 mg/kg and 20 mg/kg BW, respectively, Serumwerk Bernburg) and perfused intra-cardially with 1x PBS followed by 4% paraformaldehyde (PFA) in PBS. Brains were post-fixed for an additional 24 hr in 4% PFA at 4°C. Brains were embedded in agarose (3% in Water) and 70 μm coronal sections were cut with a VT1000S vibratome (Leica Biosystems). Every second section, ranging between approximately +2.65 to −6.2 mm from Bregma, was

mounted on glass slides using a custom-made mounting medium containing Mowiol 4–88 (Roth, Germany) as described elsewhere (*Mowiol mounting medium, 2006*) with 0.2 mg/mL DAPI (Sigma-Aldrich, MO).

## Imaging

Slides containing rabies virus tracings were imaged using a 5x/0.15 NA objective on an Axioplan2 epifluorescent microscope (Zeiss, Jena, Germany) equipped with a Ludl controllable stage (Visitron Systems, Puchheim, Germany), a CoolSnapHQ$^2$ CCD camera (Teledyne Photometrics, AZ), and orchestrated by µManager 2.0 beta software (*Edelstein et al., 2014*). Excitation was provided by an X-cite halogen lamp (Excelitas Technologies, MA) with 350/50x (DAPI) and 470/40x (eGFP) filter cubes.

Axonal AAV tracings were imaged on an SP5 or SP8 laser scanning confocal microscope (Leica, LAS AF and LAS X 3.5.0.18371, respectively) using a 10x/0.40 NA objective, and a 1 Airy disc pin-hole. 405 nm and 488 nm laser lines were used to image DAPI and eYFP channels. Single optical z-section images of 10 µm thickness from the middle (z-axis) of the section were acquired. For each brain, we determined the densest efferents outside the insular cortex, and adjusted the acquisition settings to obtain a nearly saturated signal.

Starter volumes for RV tracings were determined by imaging sections covering the injection site with an SP5 microscope using the 10x objective. 10 z-stacks of 7 µm step-size through each section were acquired. For AAV starter cells, sections covering the injection site were imaged as a single plane on the epifluorescent microscope with a 5x objective.

## Starter volume detection

Both RV and AAV starter cell volumes were determined semi-automatically using CellProfiler 3.0.0 (*Kamentsky et al., 2011*). For each image, a set of ROIs were defined for the insular and adjacent regions present. For RV images, rabies-virus-positive cells were detected in the eGFP image, and the corresponding cell objects masked over the mCherry (TVA) image. mCherry signal was then detected and back-related to the eGFP$^+$ cell. The individual double-positive cells were traced through the z-stacks and related to their corresponding ROI. For AAV images, eYFP$^+$ cell bodies were segmented and related to their corresponding ROI.

To determine starter cell volumes per cortical layer, ROIs were defined for the layers and cells counted using the Cell Counter plugin in FIJI (Fiji is just ImageJ, NIH). The option 'show all' was active, so no cells were counted more than once. For RV tracings, only double-positive cells within the insular were counted throughout the z-stacks. For AAV tracings, eYFP$^+$ cells in the insular throughout the z-stacks were counted.

To address the potential issue that starter cell spillover into brain regions neighboring the IC could affect our data, we compared starter cell distributions between all three brains of each condition. We quantified the percentage of spillover into the piriform cortex (Pir) and motor and sensory cortex (M/S). We then compared the connectivity patterns of the brains with the least amount/absent spillover and the brain with the highest amount of spillover. Only brains that did not show differences to brains without spillover were included in this study. Overall, we excluded two brains where starter cell populations were clearly detected outside the IC and that would have shown different connectivity patterns, and three brains that did not yield strong starter cell populations.

## Monosynaptic retrograde rabies virus tracing

All image processings were performed in FIJI (*Figure 1—figure supplement 4A*). Collated images for each brain section were stitched to a single image with the Grid/StitchCollection plugin. Autonomous detection of labeled neurons was performed using a customized macro script. eGFP images were background subtracted (rolling ball, pixel width 20), and the eGFP$^+$ cell bodies detected using Trainable Weka Segmentation (University of Waikato, New Zealand), trained on a small subset of images for each tracing. For training of the classifier, images from a single brain with different amounts of eGFP+ cell bodies and noise were used. After background subtraction, the Trainable Weka Segmentation classifier was retrained until it reliably picked up the cell bodies in the subsets of images used for training. The settings obtained through this process were applied for autonomous eGFP+ neuron detection in which segmented images were binarized and a watershed

segmentation run (https://github.com/GogollaLab/tracing_quantification_and_analysis/blob/master/autonomous_neuron_detection.ijm). To count labeled neurons and assign them to a brain region, a second customized macro script was used on the binary image (https://github.com/GogollaLab/tracing_quantification_and_analysis/blob/master/counting_RV.ijm).

'We created our own ROI-sets for each brain section by merging information from coronal maps of two mouse reference atlases (Paxinos and Franklin, and Allen Brain Atlas). The library can be found here: https://github.com/GogollaLab/tracing_quantification_and_analysis/tree/master/ROI_set_atlas.

For each section to analyze, we first determined the distance from Bregma, taking into account previous and following sections to achieve the most accurate result. Then, the corresponding library ROI-set was laid over the image and moved manually to fit the section. Regions were further manually adjusted to correct for section warping, tissue damage and to exclude artefacts. The two mouse brain atlases were used as references to ensure proper alignment. In case of uneven cutting, more than one reference ROI-set was used.

The number of positive cells was determined using the 'Analyze Particles' plugin (size = 70–1000, circularity = 0.30–1.0). Data output was calculated as cell counts for a given ROI normalized to the total cell counts for the individual brain (% of total input). Additionally, cell density was calculated as total cell number per ROI area. The injection site was excluded from the analysis, to ensure no starter cells are counted as input cells.

We calculated the relative difference between three individual human counts as:

$$RPD_1 = \left| \frac{(human\ count1 - human\ count2 - human\ count3)}{(human\ count1 + human\ count2 + human\ count3)/3} \right|$$

We tested the performance of our semiautomated approach compared to the average human counts by:

$$RPD_2 = \left| \frac{(average\ human\ count - automated\ count)}{(average\ human\ count + automated\ count)/2} \right|$$

We compared the counts of one human in three brains versus the automated approach using the same equation as $RPD_2$.

## Axonal AAV tracing

Collated images were stitched for each brain section using Leica Application Suite X 3.3.0.16799. Image processing was done in FIJI using customized macro scripts (*Figure 1—figure supplement 4B*). First, hessian ridge detection and thresholding was performed as described elsewhere (*Grider et al., 2006*). Threshold settings were determined before running the macro script to ensure equal processing within a sample. Briefly, this results in binary images of the eYFP$^+$ axons while eliminating background fluorescence (https://github.com/GogollaLab/tracing_quantification_and_analysis/blob/master/autonomous_pixel_detection.ijm). These images were then quantified with a second script where, similar to the rabies virus quantification, the custom-made ROI atlas was manually adjusted for every coronal section (https://github.com/GogollaLab/tracing_quantification_and_analysis/blob/master/counting_AAV.ijm). The particle analyzer was used to count pixels. Percent of total output was calculated from the thresholded image, with the eYFP$^+$ pixel count of each ROI normalized to the total of all eYFP$^+$ pixels identified from the individual brain. Additionally, percent innervation density was calculated as the proportion of eYFP$^+$ pixels covering the maximal pixel count for its ROI. Clearly distinguishable passing fiber bundles (such as in the striatum, cerebral peduncles, anterior commissure, internal- and external capsules, and pyramidal tract) were excluded from the analysis. As with the RV tracings, the starter volume was also excluded from all analysis.

## Data collection and statistical analysis

Data acquired by RV and AAV tracings was analyzed using a custom-written code in Python 3.6 (see *Figure 1—figure supplement 4C* and the code found at: https://github.com/GogollaLab/tracing_quantification_and_analysis/blob/master/analysis_RV.py, https://github.com/GogollaLab/tracing_quantification_and_analysis/blob/master/analysis_AAV.py). Cells (for RV) and pixels (for AAV) were grouped in both 17 large brain regions, and the 75 sub-regions thereof. Regions with less than

0.03% connectivity were considered below background threshold, and set to zero. Starter cell volume and artificial signals were excluded before further processing of data. First, total cell count across bregma/ROI and ROI/bregma were calculated. After calculation of cell count, percent input and density/bregma, data was clustered into lower and higher hierarchies. Separate analysis of amygdala, thalamus and striatum was obtained by sorting of the respective ROIs into separate data frames for the creation of pivot tables.

To create plots that display the data along the anterior-posterior axis (e.g. % density innervation), we first linearly interpolated missing values and then smoothed the data using a Savitzky-Golay-Filter (scipy.signal.savgol_filter).

Group comparison of connectivity patterns were made using ordinary one-way ANOVAs per subregion, followed by Tuckey's multiple comparisons test. The same test was used to compare starter cell distribution over regions between injection sites (aIC, mIC, pIC). Comparisons between starter cell distributions over cortical layers were done with a two-way RM ANOVA followed by Tuckey's multiple comparisons test. For input-output correlations to test for reciprocity, analysis was performed using GraphPad Prism (GraphPad Software, CA). For the correlation matrices of input vs. input and output vs. output, the data of the 17 major brain regions (% of total in- or output) was correlated by computing the pair-wise Pearson's correlation coefficients of all input or output tracings, respectively. Then, the correlation coefficients were hierarchically clustered with the complete-linkage clustering method. All quantifications of RV and AAV tracings were done masked.

All codes for quantification and analysis of RV and AAV tracings are available at https://github.com/GogollaLab/tracing_quantification_and_analysis.

All animal numbers are reported in Figures and their legends. No statistical methods were used to predetermine sample size, but it is comparable to published work (*Ährlund-Richter et al., 2019*; *Do et al., 2016*; *Luo et al., 2019*).

## Acknowledgements

We thank M Junghänel, F Lyonnaz, M Ponserre for technical assistance and N Dolensek for helping to set up the image acquisition pipeline. This study was supported by the Max Planck Society, the Deutsche Forschungsgemeinschaft (SPP1665 to K-KC, DAG and NG), funding from the European Research Council (ERC) under the European Union's Horizon 2020 research and innovation programme (ERC-2017-STG, grant agreement 758448 to NG), and the ANR-DFG project 'SAFENET' (ANR-17-CE37-0021 to AK and NG).

## Additional information

### Funding

| Funder | Grant reference number | Author |
| --- | --- | --- |
| Max-Planck-Gesellschaft | | Caroline Weiand<br>Nadine Gogolla |
| Deutsche Forschungsgemeinschaft | SPP1665 | Daniel August Gehrlach<br>Alexandru A Hennrich<br>Karl-Klaus Conzelmann<br>Nadine Gogolla |
| Horizon 2020 Framework Programme | ERC-2017-STG 758448 | Thomas N Gaitanos<br>Nadine Gogolla |
| Agence Nationale de la Recherche | ANR-17-CE37-0021 | Alexandra S Klein<br>Nadine Gogolla |

The funders had no role in study design, data collection and interpretation, or the decision to submit the work for publication.

### Author contributions

Daniel A Gehrlach, Conceptualization, Data curation, Software, Formal analysis, Supervision, Validation, Investigation, Visualization, Methodology, Writing - original draft; Caroline Weiand, Data

curation, Formal analysis, Validation, Investigation, Project administration, Writing - review and editing; Thomas N Gaitanos, Data curation, Formal analysis, Investigation, Visualization, Writing - original draft; Eunjae Cho, Formal analysis; Alexandra S Klein, Investigation; Alexandru A Hennrich, Resources; Karl-Klaus Conzelmann, Conceptualization, Resources; Nadine Gogolla, Conceptualization, Supervision, Funding acquisition, Writing - original draft, Project administration, Writing - review and editing

## Author ORCIDs
Caroline Weiand (ID) https://orcid.org/0000-0001-8497-1282
Nadine Gogolla (ID) https://orcid.org/0000-0003-3754-7133

## Ethics
Animal experimentation: All animals were used in accordance with the regulations and under licenses obtained from the government of Upper Bavaria (Animal license AZ: 55.2-1-54-2532-56-2014).

## Decision letter and Author response
Decision letter https://doi.org/10.7554/eLife.55585.sa1
Author response https://doi.org/10.7554/eLife.55585.sa2

# Additional files

## Supplementary files
- Supplementary file 1. List of abbreviations.
- Supplementary file 2. Entire dataset for pivot tables.
- Supplementary file 3. Detailed statistics.
- Transparent reporting form

## Data availability
All data generated or analysed during this study are included in the manuscript and supporting files. Source data files have been provided in Supplementary file 2.

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
