## [Decision Letter]

**Acceptance summary:**

The role of the insula cortex is widely varied and highly complex. There remain questions regarding the connectivity of its global inputs and outputs. This manuscript presents a comprehensive dataset of insular cortex anterograde and retrograde mapping using modern tools such as AAVs, monosynaptic rabies tracing, and high-quality quantitative analyses. The study is technically very well executed. Semi-automated approaches for brain alignment and cell quantification to reduce bias are used and discussed. While some of the results presented have been previously reported in earlier studies, presenting it all in one place within a consistent experimental and analytical framework is extremely useful for advancing future investigations of insular cortex. The data is of high quality and is plotted in ways that make it easy for the reviewer to digest these large data sets.

**Decision letter after peer review:**

Thank you for submitting your article "A whole-brain connectivity map of mouse insular cortex" for consideration by *eLife*. Your article has been reviewed by 3 peer reviewers, and the evaluation has been overseen by Kate Wassum as the Senior Editor and Reviewing Editor. The following individual involved in review of your submission has agreed to reveal their identity: Yoav Livneh (Reviewer #1).

The reviewers have discussed the reviews with one another and the Reviewing Editor has drafted this decision to help you prepare a revised submission.

Essential revisions:

1) Statistical comparison of the connectivity patterns in Figures 2-5 need to be run. With many regions and a small n (n=3), it is likely that many of the differences will not be significant after correcting for multiple comparisons. Thus we suggest to increase n to ensure sufficient statistical power to make more concrete claims about the organization of connectivity. This will help support some of the main conclusions.

2) Since Cre drivers were broadly expressed, please report which layers the starter cells occupied. Though they are claimed to target all layers, in the representative images shown in Figure 1—figure supplement 1A and 1B, the majority of starter cells seem to be in layer 5. Adding n may also allow them to identify biases in input/output organization based on the layer distribution of the starter cells.

3) Rabies traced cells tend to have highly variable fluorescence levels and machine thresholding tends to exclude many dim cells. Please include human counts for several rabies brains to verify the cell counts collected by the algorithm. Using several human counts to verify the rabies cell numbers in a single brain, please verify the expected percent error for various human counts and machine counts.

4) There are several places where additional details and clarity are needed:

a) Aligning atlas borders to coronal sections may not accurately define area borders, especially if there are defects in the tissue. Warping or skewing due to mounting and shrinking, as well as imperfect cutting often lead to coronal sections that do not match the atlas. How are reliable landmarks being used across the entire surface of the tissue to properly warp the atlas borders to the tissue? Please clarify.

b) Regarding spillover to Pir, S1, S2 and M1 – the authors mention that they "…asked if these contaminations affected the qualitative connectivity structure by comparing them to tracings without contamination". However, it is unclear how this was done and no data is presented to demonstrate this. Please describe this in detail and present examples of excluded data and criteria for exclusion? This is a critical point for interpretation of the data.

c) When detecting axons, how was thickness handled? Depending on the pixel resolution, it is possible that bundled or single axons will not be represented proportionally. In addition, differing saturations can result in halo effects and thickening. Were these types of effects considered when counting pixels?

d) In Figure 1 and main text, the author described "minimal (or small) percentage of cells” were detected in the M/S, Pir. Please indicate the actual numbers. Any starter neurons in other IC regions? Thus, in mIC or pIC when aIC was targeted etc.

e) The authors should provide more details about their analysis methods, especially for the machine learning-based approaches.

f) Overall the methods about quantification of axon tracing should be described more in detail. Specifically, the steps applied in the custom FIJI macros used.

5) Axon pixel counts rely on methods that are highly sample dependent as noted by Grider et al., 2006. Grider et al. maintained uniform acquisition settings across the samples, however, variable acquisition settings were used by the authors. In addition, thresholding in this manuscript was done for each image, a destructive task, and hessian ridge detection can then find more or fewer pixels than existed in the original image according to Grider. These results may be more qualitative than quantitative, on the contrary to what the authors described in their main text. Please clarify and address this.

[Editors' note: further revisions were suggested prior to acceptance, as described below.]

Thank you for resubmitting your work entitled "A whole-brain connectivity map of mouse insular cortex" for further consideration by *eLife*. Your revised article has been evaluated by Kate Wassum (Senior Editor) and a Reviewing Editor.

The manuscript has been improved but there are some remaining issues that need to be addressed before acceptance, as outlined below:

Essential revisions:

1) In this revised manuscript, the authors ran multiple t-tests instead of increasing number n. Multiple t-tests are not the best test since they increase the Type I error rate. An ANOVA with posthoc multiple-comparison tests, or non-parametric to be more meticulous, would be more appropriate. Please include this analysis.

2) Contrary to the authors' argument using RPDs, human count versus automatic counting do not correspond well. For example, according to Supplementary file 1, sheet 2, B section, in aIC-Po, human 3 = 26, automated = 133. This is an order of magnitude difference. In pIC-Po, human 3 = 249, automated = 118. Then, this time the human 3 counts more than ~100 cells than the automated does. Thus, the trends are opposite. In addition, RPD cannot predict the directionality; thus, whether the humans tend to count more or less than the automated counting does. This concern should be addressed. The reviewers suggested that showing the raw data overlaid with automatically identified cells would be a helpful first start to understand the sources of discrepancies and that perhaps the classifier needs to be tweaked for different brain regions which have different densities of cells labeled.

3) Figure 1—figure supplement 2: In aIC input excitatory tracing, 20% of starter neurons were found outside of IC, thus Pir. This is a significant number. The difference between aIC and mIC/pIC must result from Pir inclusion? What are the results of Pir input tracing? Or by using a simple retrograde tracer injection (cholera toxin subunit B) to Pir, which regions will be labeled? Would that affect aIC input tracing results? It is not clear to me whether the paired t-test is the right choice of analysis to answer the contaminated starting neuron question. Please also address this remaining concern. Reviewers suggested that adding more animals with better targeting would be the best solution to this problem. If that's not possible, comparing the results to existing data for PIR, e.g., from the online resources, would be helpful. If neither of these can address the issues, it would strengthen the conclusion to use more stringent spillover exclusion criteria.

---

## [Author Response]

Essential revisions:1) Statistical comparison of the connectivity patterns in Figures 2-5 need to be run. With many regions and a small n (n=3), it is likely that many of the differences will not be significant after correcting for multiple comparisons. Thus we suggest to increase n to ensure sufficient statistical power to make more concrete claims about the organization of connectivity. This will help support some of the main conclusions.

We thank the reviewers for their comment. As agreed in our correspondence with the Senior and Reviewing Editor, Kate Wassum, in light of the disruption to research as a result of the pandemic, we have now performed statistical tests on the existing data (n=3) but without increasing the n size.

We ran multiple t-tests and corrected for multiple comparisons using Holm-Sidak. We performed these statistical comparisons between subregions of the IC, and also between excitatory and inhibitory inputs to different subregions. We amended the steps of analysis to the Materials and methods and updated the Figures 2-5 as well as Figure 2—figure supplements 1-3, including all corresponding figure legends. In addition, we updated the main text to include these results and provide a detailed account of all statistics in Supplementary file 3.

2) Since Cre drivers were broadly expressed, please report which layers the starter cells occupied. Though they are claimed to target all layers, in the representative images shown in Figure 1—figure supplement 1A and 1B, the majority of starter cells seem to be in layer 5. Adding n may also allow them to identify biases in input/output organization based on the layer distribution of the starter cells.

We thank the reviewers for this valid criticism. To address this question, we quantified the layer distribution of starter cells. The detailed steps of analysis are included in the Materials and methods section. This analysis revealed that starter cells were found in all cortical layers but as highlighted by the reviewers in different proportions. While we did find differences in distribution of starter cells between layers, our analysis revealed that these differences are not likely influenced by the Cre-lines or the viruses we used, since the same ratios of starter cell distributions were found in GAD2-Cre and CamKII-Cre lines, but also for AAV and RV tracings. We consistently found most starter cells in layers III and V, less in II and VI and least in I and IV. We now included this important information about starter cell distributions in the Results and in Figure 1—figure supplement 2B.

3) Rabies traced cells tend to have highly variable fluorescence levels and machine thresholding tends to exclude many dim cells. Please include human counts for several rabies brains to verify the cell counts collected by the algorithm. Using several human counts to verify the rabies cell numbers in a single brain, please verify the expected percent error for various human counts and machine counts.

To address this concern, we conducted several human counts for three representative brain regions. We chose to analyze subregions of the amygdala, the thalamus and the striatum for different conditions (aIc, mIC and pIC), as we discuss these regions in length in the manuscript. Additionally, we used three independent human counts in one single brain, which was traced from the aIC.

Since we do not have a true value for the cell counts, we first calculated the relative percent difference (RPD) between several human counts using the following equation:

RPD_1_ =│ (humancount1−humancount2–humancount3)(humancount1+humancount2+humancount3)/3 │

We found that between humans the RPD_1_ = 1.35 %.

We next generated an RPD to estimate the relative difference between average human and our automated method. Towards this goal, we compared the average human counts (for 3 human individuals) to the automated count using the following equation:

RPD_2_ = │averagehumancount–automatedcount(averagehumancount+automatedcount)/2)│

We found the RPD_2_ = 0.54 %.

Finally, we compared for three different brains human versus automated counts and found a low RPD of 0.29.

In sum, these analyses argue that our automated method was in general comparable to human counts and may even be more consistent. Please refer to Supplementary file 1 for detailed results and raw counts and the Results for the mentioning of this validation.

4) There are several places where additional details and clarity are needed:a) Aligning atlas borders to coronal sections may not accurately define area borders, especially if there are defects in the tissue. Warping or skewing due to mounting and shrinking, as well as imperfect cutting often lead to coronal sections that do not match the atlas. How are reliable landmarks being used across the entire surface of the tissue to properly warp the atlas borders to the tissue? Please clarify.

We understand the concern of the reviewers, as brain sections are indeed often prone to damage and warping. We amended the Materials and methods section to explain in detail how on top of automated alignment we added manual adjustments for each ROI:

“We created our own ROI-sets for each brain section by merging information from coronal maps of two mouse reference atlases (Paxinos and Franklin, and Allen Brain Atlas). The library can be found here:

https://github.com/GogollaLab/tracing_quantification_and_analysis/tree/master/ROI_set_atlas

For each section to analyze, we first determined the distance from Bregma, taking into account previous and following sections to achieve the most accurate result. Then, the corresponding library ROI-set was laid over the image and moved manually to fit the section. Regions were further manually adjusted to correct for section warping, tissue damage and to exclude artefacts. The two mouse brain atlases were used as references to ensure proper alignment. In case of uneven cutting, more than one reference ROI-set (e.g. of subsequent coronal sections) was used.”

b) Regarding spillover to Pir, S1, S2 and M1 – the authors mention that they "…asked if these contaminations affected the qualitative connectivity structure by comparing them to tracings without contamination". However, it is unclear how this was done and no data is presented to demonstrate this. Please describe this in detail and present examples of excluded data and criteria for exclusion? This is a critical point for interpretation of the data.

We agree to this valid criticism and apologize for the unclear formulation of our exclusion criteria. To address the potential issue of spillover, we compared starter cell distributions between all three brains of each condition. We quantified the percentage of spillover into the piriform cortex (Pir) and motor and sensory cortex (M/S). These data are now plotted in Figure 1—figure supplement 2C. We then compared the connectivity patterns of the brains with the least amount/absent spillover and the brain with the highest amount of spillover (please refer to Figure 1—figure supplement 2D). In order to quantitatively assess whether differences are explained by different amounts of spillover, we performed paired t-tests and found no significant connectivity differences between brains exhibiting different amounts of spillover but belonging to the same tracing condition.

We excluded two brains where starter cell populations were clearly detected outside the IC (see example picture Figure 1—figure supplement 2A, right) and that would have showed different connectivity patterns, and three brains that did not yield strong starter cell populations. We now clearly describe the exclusion criteria in the main text and the Materials and methods section. We also included representative images of included and excluded starter cell distributions in Figure 1—figure supplement 2A.

c) When detecting axons, how was thickness handled? Depending on the pixel resolution, it is possible that bundled or single axons will not be represented proportionally. In addition, differing saturations can result in halo effects and thickening. Were these types of effects considered when counting pixels?

It is true that the virus and the analysis method we employed do not allow us to differentiate between single axons and very dense axon bundles. We thus had to rely on quantifying eYFP-positive pixels above the threshold instead of numbers of individual axons. Consequently, also axon thickness could not be taken into account. Since the pixels of halo effects are lighter than those of axons or dense innervations these effects were excluded in the step of thresholding. We apologize for not stating this clearly enough. You can now find a clarification in the main text.

d) In Figure 1 and main text, the author described "minimal (or small) percentage of cells” were detected in the M/S, Pir. Please indicate the actual numbers. Any starter neurons in other IC regions? Thus, in mIC or pIC when aIC was targeted etc.

Please note that we have now included an additional figure supplement (Figure 1—figure supplement 2) in which percentage of spillover into M/S and Pir is quantified. Further, we would like to bring to the reviewers’ attention that spillover into other IC regions can be found in Figure 1C as well as in Figure 1—figure supplement 1C. For a more detailed response about spillover, please refer to our answer for question 4b.

e) The authors should provide more details about their analysis methods, especially for the machine learning-based approaches.

We apologize for the lack of detailed methods. In order to make our quantification and analysis methods more comprehensive for the readers we revised the Materials and methods section. Additionally, we now include workflow diagrams in Figure 1—figure supplement 3 displaying the analysis in an approachable manner. We also included links to the corresponding FIJI macros and python scripts, which we made available on GitHub.

f) Overall the methods about quantification of axon tracing should be described more in detail. Specifically, the steps applied in the custom FIJI macros used.

Please see our answer to the comment above. We provide additional information in the Materials and methods section, including corresponding links to GitHub as well as the workflow diagram Figure 1—figure supplement 3B, C.

5) Axon pixel counts rely on methods that are highly sample dependent as noted by Grider et al., 2006. Grider et al. maintained uniform acquisition settings across the samples, however, variable acquisition settings were used by the authors. In addition, thresholding in this manuscript was done for each image, a destructive task, and hessian ridge detection can then find more or fewer pixels than existed in the original image according to Grider. These results may be more qualitative than quantitative, on the contrary to what the authors described in their main text. Please clarify and address this.

We agree with the reviewers that results of AAV pixel counts are more qualitative than quantitative. Therefore, we used average of the percent of total output per each brain when comparing between conditions. However, given the fact that we had a variety of different conditions from different starting points to analyze, we had to adapt the protocol used by Grider et al. to be compatible with our tracing data. Specifically acquisition and thresholding had to be adjusted to our needs, because our samples had different sizes of starter cell populations, different durations of expression and were acquired from three different subregions of the Insula Cortex (aIC, mIC, pIC). In order to be able to use the same acquisition settings over all brains, it would have been necessary to image all of the brains to identify the fluorescence histogram before acquiring the images for quantification. However, uniform settings could have compromised the quality of the acquired images.

Furthermore, also within a single brain, acquisition settings had to be balanced to ensure that all positive signals were picked up without overexposing densely innervated regions. To achieve this, the most densely innervated region outside the injection site was placed at the end of the dynamic range (see Materials and methods). Despite those limitations, acquisition settings between samples were still highly similar.

Since our aim was to count pixels instead of axons, hessian ridge detection and subsequent thresholding is a necessary step to obtain a binary image. Since these processes were automated for the whole brain, thresholding was performed on every image separately. Nevertheless, threshold settings did not change within a brain, so that there was no difference in treatment of single sections within a sample.

We now extended the details given in the Materials and methods section concerning the acquisition of images and quantification of pixels. Additionally, we now clearly state this circumstance in the limitations. We further now included a comprehensive workflow diagram detailing the individual steps applied in FIJI (Figure 1—figure supplement 3B).

[Editors' note: further revisions were suggested prior to acceptance, as described below.]

Essential revisions1) In this revised manuscript, the authors ran multiple t-tests instead of increasing number n. Multiple t-tests are not the best test since they increase the Type I error rate. An ANOVA with posthoc multiple-comparison tests, or non-parametric to be more meticulous, would be more appropriate. Please include this analysis.

As suggested by the reviewer, we have now performed a one-way ANOVA followed by Tukey’s multiple comparison test for each brain region as done in other comparable studies (Beier et al., 2015; Luo et al., 2019; Wang et al., 2019).

Most of the major conclusions remain unaltered but we did not detect any significant differences between inhibitory and excitatory neuron connectivity any longer (of which there were very few using unpaired t tests). However, as we already argued in our first rebuttal, we consider our study and the tracing techniques we employ, in the absence of physiological measurements, as more qualitative than quantitative in nature. This notion is also reflected in many studies in the field that do not quantitatively compare their results (Do et al., 2016; Su et al., 2018; Watabe-Uchida et al., 2012). We discuss these limitations in detail in the Discussion.

In sum, we hope that performing the one-way ANOVA and discussing the limitations does satisfy the reviewers concern.

2) Contrary to the authors' argument using RPDs, human count versus automatic counting do not correspond well. For example, according to Supplementary file 1, sheet 2, B section, in aIC-Po, human 3 = 26, automated = 133. This is an order of magnitude difference. In pIC-Po, human 3 = 249, automated = 118. Then, this time the human 3 counts more than ~100 cells than the automated does. Thus, the trends are opposite. In addition, RPD cannot predict the directionality; thus, whether the humans tend to count more or less than the automated counting does. This concern should be addressed. The reviewers suggested that showing the raw data overlaid with automatically identified cells would be a helpful first start to understand the sources of discrepancies and that perhaps the classifier needs to be tweaked for different brain regions which have different densities of cells labeled.

We agree with the reviewers that our manual counts compared very poorly to the automated counts, which puts our automated assessment into question. We have thus revisited this issue thoroughly.

First, we checked the automatically detected cells. To allow the reviewers and readers to judge the performance of the automated algorithm, we now included example pictures for several brain regions depicting the fluorescent signal, as well as the cells detected by a human and the automated cells counts on top of the segmentation (as can be seen in Figure 1—figure supplement 3).We believe that the automated counts reflect a very objective and truthful representation of the data.

Secondly, we tried to understand the source of error in our previously reported human counts. Since we realized that our human counters reported input cells from the CeA to the aIC, which does not reflect the reality and human counts varied greatly for the Po, we tried to understand the source of these variations or misinterpretations by the human counters. Indeed, we found that the human counters, which were students from our lab that unfortunately did not have much prior experience in quantifying rabies tracings, were not consistently including or excluding signals that were either too small to be considered as true input cells or did inconsistently include or exclude cells that lay on top of the ROI borders. Since counting was done over several sections for every region, even small mistakes added up, resulting in an overall large difference. We apologize for this oversight in properly instructing the human counters. Unfortunately, due to the lockdown, we had to instruct human counters remotely, which added to the potential for mistakes. We now carefully instructed counters on common criteria before the counting and report new cell counts for all previous regions. We also expanded this analysis to three additional brain regions. Our goal was to assess variations between human and automated counts for regions that have many or very few input cells.

We now report total counts/brain region, which are more consistent between humans and for humans versus automated counts. Furthermore, we report the discrepancies per brain section since the counts per brain region were assessed over multiple sections increasing the likelihood of error. This analysis reveals that per individual brain section the differences between various counts was very small.

In conclusion, we hope that these analyses convince the reviewers and readers of the validity of our quantification approach. The example images, new counts and RPDs are included in Figure 1—figure supplement 3.

3) Figure 1—figure supplement 2: In aIC input excitatory tracing, 20% of starter neurons were found outside of IC, thus Pir. This is a significant number. The difference between aIC and mIC/pIC must result from Pir inclusion? What are the results of Pir input tracing? Or by using a simple retrograde tracer injection (cholera toxin subunit B) to Pir, which regions will be labeled? Would that affect aIC input tracing results? It is not clear to me whether the paired t-test is the right choice of analysis to answer the contaminated starting neuron question. Please also address this remaining concern. Reviewers suggested that adding more animals with better targeting would be the best solution to this problem. If that's not possible, comparing the results to existing data for PIR, e.g., from the online resources, would be helpful. If neither of these can address the issues, it would strengthen the conclusion to use more stringent spillover exclusion criteria.

We agree with the reviewer that specifically the rabies tracing from excitatory cells in the aIC suffer from a large Pir contamination. The reviewer raises the valid concern that this contamination may explain the differences we report for aIC versus mIC and pIC connectivity.

We thus compared region-by-region which inputs were reported as different for the aIC compared to mIC/pIC:

1) In Figure 2A, top, we observe a difference between sensory cortex inputs to the pIC compared to mIC and aIC (only a trend and not significant). Since we observe more inputs for pIC than mIC or aIC, and the difference establishes already between the pIC and mIC, we can conclude that this difference is not affected by the Pir contamination.

2) All other cortical inputs reported in Figure 2A, top, seem to target the aIC, mIC and pIC rather homogeneously and only non-significant trends, such as for example for motor cortex, are reported.

3) In Figure 2A, bottom, we report subcortical input to the different insular subdivisions. We here detect a significantly stronger innervation of the mIC compared to the aIC from olfactory areas and a trend towards the same for the amygdala. Since the innervation is stronger for mIC, it cannot be attributed to a Pir contamination in aIC tracings. The stronger olfactory area connectivity may also relativize the actual contribution of the Pir contamination to our overall aIC versus mIC results, since one would expect a stronger olfactory connectivity if the aIC tracings were indeed strongly biased by the starter cell in the Pir. Finally, also the inhibitory neurons in the aIC and mIC exhibit the same difference. However, the inhibitory tracings did not suffer from contamination, which makes us confident that also the reported differences are true.

4) Again in Figure 2A, bottom, the only global brain region, which seems to shows a stronger input connectivity between aIC, and mIC is the thalamus. We thus looked into more details of the thalamic connectivity to the aIC versus mIC and pIC as shown in Figure 5. While we detected few differences between the aIC and the mIC/pIC, only for one thalamic region was the input connectivity stronger from the thalamus to the aIC and could thus be influenced potentially by the Pir contamination. This difference was true for the CM part of the intralaminar nuclei of the dorsal thalamus. While we cannot formally exclude that this strong difference in connectivity was caused by Pir contamination, we believe that this is unlikely since the inhibitory neuron connectivity shows the exact same pattern yet does not suffer from Pir contamination and we find that overall the excitatory and inhibitory connectivity in the insular cortex is very balanced. Furthermore, this particular nucleus was not described as a major input region of the Pirifrom cortex itself in a very comprehensive input tracing study of cell-type specific inputs to the anterior and posterior pirifrom cortex (Wang et al., 2020). Instead, this study highlights a strong connectivity of the anterior pirifrom cortex with the mediodorsal thalamus. While we do see strong aIC connectivity to the MD, the connectivity is not significantly stronger than to mIC and thus also here, the Pir contamination is not a driver of major differences. Furthermore, it is well established that the MD sends major inputs to the aIC in rats and mice (Allen et al., 1991; Mátyás et al., 2014; Shi and Cassell, 1998).

5) Additionally, our conclusion about the differences for aIC versus mIC/pIC connectivity is also based on the inhibitory connectivity patterns which do not suffer from a Pir contamination.

Taken together, we thus believe that while we do have a sizeable contamination of starter cells lying in the Pir in our rabies tracing to excitatory cells of the aIC, based on the nature of differences observed and the existing literature, this contamination does not seem to change the main conclusions on aIC connectivity we make in this study.

We now discuss this limitations in more detail in the Discussion.